# Mass spectrometry uncovers intermediates and off-pathway complexes for SNARE complex assembly

Julia Hesselbarth[1,2] & Carla Schmidt [1,2✉]

The SNARE complex assembles from vesicular Synaptobrevin-2 as well as Syntaxin-1 and SNAP25 both anchored to the presynaptic membrane. It mediates fusion of synaptic vesicles with the presynaptic plasma membrane resulting in exocytosis of neurotransmitters. While the general sequence of SNARE complex formation is well-established, our knowledge on possible intermediates and stable off-pathway complexes is incomplete. We, therefore, follow the stepwise assembly of the SNARE complex and target individual SNAREs, binary sub-complexes, the ternary SNARE complex as well as interactions with Complexin-1. Using native mass spectrometry, we identify the stoichiometry of sub-complexes and monitor oligomerisation of various assemblies. Importantly, we find that interactions with Complexin-1 reduce multimerisation of the ternary SNARE complex. Chemical cross-linking provides detailed insights into these interactions suggesting a role for membrane fusion. In summary, we unravel the stoichiometry of intermediates and off-pathway complexes and compile a road map of SNARE complex assembly including regulation by Complexin-1.

[1] Interdisciplinary Research Centre HALOmem, Charles Tanford Protein Centre, Institute of Biochemistry and Biotechnology, Martin Luther University Halle-Wittenberg, Halle, Germany. [2] Present address: Department of Chemistry – Biochemistry, Biocenter II, Johannes Gutenberg University Mainz, Mainz, Germany. ✉email: carla.schmidt@uni-mainz.de

Signal transmission in neurons is accomplished by neurotransmitter release from the lumen of synaptic vesicles into the synaptic cleft of synapses where they are received by postsynaptic receptors[1]. To achieve this, the neuronal SNARE (i.e., soluble N-ethylmaleimide-sensitive factor attachment protein receptor) complex brings the vesicle and presynaptic membranes in close proximity, thereby, mediating the fusion of the two membranes resulting in exocytosis of neurotransmitters. The SNARE complex assembles from the vesicular protein Synaptobrevin-2 as well as SNAP25 (synaptosome-associated protein of 25 kDa) and Syntaxin-1, which are both anchored to the presynaptic membrane[2]. While individual SNARE proteins are unstructured, the presence of other SNAREs induces conformational changes leading to the formation of a stable and twisted bundle of four parallel alpha-helices[3–5]. Zippering of the SNAREs is proposed to provide the required energy to initiate membrane fusion[6,7] and proceeds from the N-terminus towards the membrane-proximal C-terminus through interactions of their complementary SNARE motifs. The SNARE motifs are 60-70 amino acids long and are conserved in Synaptobrevin-2, Syntaxin-1 and SNAP25; the latter contains two SNARE motifs. Accordingly, Syntaxin-1 and Synaptobrevin-2 each contribute one and SNAP25 contributes two alpha-helices to the ternary SNARE complex. The assembled helical bundle contains 15 parallel layers of hydrophobic amino acids and one central ionic layer, the so-called zero layer, which is composed of three highly conserved glutamine and one arginine residues[5]. Based on the presence of glutamine or arginine residues in the zero layer, the SNAREs are classified as Qa- (Syntaxin-1), Qb- and Qc- (SNAP25) or R-SNAREs (Synaptobrevin-2) resulting in the formation of the ternary QabcR (3Q:1R) SNARE complex[8]. In addition to the parallel assembly of the alpha-helices of the functionally active ternary SNARE complex, antiparallel configurations have been described that spontaneously assemble albeit at lower stability than the parallel four-helix bundle[9,10].

Fusion of synaptic vesicles with the presynaptic membrane occurs in the presence of the NSF (N-ethylmaleimide-sensitive-factor) / α-SNAP (α-soluble NSF attachment protein) disassembly machinery. Regulatory proteins are therefore required to maintain integrity of the SNARE complex[11]. According to the currently accepted model, Munc-18 (mammalian Unc-18) and Munc-13 (mammalian Unc-13) convert Syntaxin-1 to a closed[12] or open conformation[13,14], respectively, thereby enabling the formation of an acceptor complex containing SNAP25 and Syntaxin-1 as a binding site for Synaptobrevin-2. This intermediate complex is considered to be the starting point for SNARE complex assembly[15]. The later steps of SNARE-mediated membrane fusion are regulated by the vesicular calcium sensor Synaptotagmin-1 interacting with the assembled SNARE complex or the two membranes in a calcium-dependent manner[16] as well as a group of small cytosolic proteins termed Complexins[17,18].

Of these, Complexin-1 is a highly charged protein for which inhibiting and stimulatory roles in SNARE-mediated membrane fusion have been reported[19–21]. It is composed of an unstructured N-terminal domain, a dynamic accessory helix[22], a central helix including the SNARE complex binding motif[22,23] and a disordered C-terminal domain containing a tandem lipid binding motif[24,25]. Antiparallel binding of Complexin-1 to the SNARE core complex was described occurring through interactions of the central helix with a groove formed by Synaptobrevin-2 and Syntaxin-1[22,23]. Conformational changes and structural rearrangements of the SNARE complex as a result of Complexin-1 binding were proposed, however, are still controversially discussed[18,26–28].

While the overall sequence of SNARE-mediated membrane fusion is established, the exact underlying mechanisms and the structural details of the intermediates are still elusive. This includes the presence and stoichiometry of potential SNARE intermediates, possible off-pathway sub-complexes as well as interactions with regulatory factors such as Complexin-1. Previous studies provided insights into acceptor complexes differing in stoichiometry[29–31] as well as oligomerisation of individual SNAREs[32,33] and the fully assembled SNARE complex[34]. Here, we explore multimerisation and complex formation of individual SNAREs, binary SNARE sub-complexes and intermediates as well as the fully assembled, ternary SNARE complex. We also investigate the role of Complexin-1 at all stages of SNARE complex assembly and provide insights into SNARE complex stabilisation and regulation. Specifically, we use native mass spectrometry (MS) to overcome the difficulties in determining the stoichiometry of heterogeneous mixtures of complexes, and chemical cross-linking to identify interaction sites and protein arrangements including dynamic and unstructured protein regions.

## Results

**Individual SNAREs and Complexin-1 are unstructured in solution.** To study the assembly pathway and regulation of the SNARE complex including the formation of potential intermediates and stable off-pathway complexes, soluble constructs of the proteins were employed. These constructs are not attached to the membranes, therefore, allowing for parallel and antiparallel configurations of all components. In detail, we used a SNAP25 variant in which all cysteine residues were mutated to serine residues, Syntaxin-1A and Synaptobrevin-2 variants both lacking their transmembrane helices as well as full-length Complexin-1. SNARE proteins and Complexin-1 were purified from E. coli following an affinity-based purification protocol (see Materials and Methods for details). Purification of all proteins was verified by liquid chromatography-coupled tandem MS (LC-MS/MS) with high sequence coverages of 94.7% (SNAP25), 87.5% (Syntaxin-1A), 80% (Synaptobrevin-2) and 88.3% (Complexin-1). In addition, we also identified DnaK, an E. coli homolog of the human Hsp70 chaperone, presumably stabilizing SNARE proteins in solution (Supplementary Table 1).

We first characterised SNAP25 in the absence of interaction partners similar to a previous study on Synaptobrevin-2[33]. For this, SNAP25 was cross-linked with increasing amounts of bis(sulfosuccinimidyl) suberate (BS3) cross-linker which is reactive towards primary amines such as lysine residues or the protein's N-terminus and, to a lower extent, towards hydroxyl groups of serine, threonine and tyrosine residues[35]. Cross-linking was then evaluated by gel electrophoresis and western blot detection using a specific SNAP25 antibody. Similar to Synaptobrevin-2[33], SNAP25 multimers up to pentamers and heptamers were observed after Coomassie staining or Western Blot detection, respectively (Fig. 1a, Supplementary Figure 1a).

For identification of protein interactions in these multimers, 10 μM of the protein were cross-linked with a 150-fold molar excess of BS3. The cross-linked protein was then hydrolysed with trypsin generating a mixture of linear peptides and cross-linked peptide pairs. Cross-linked peptide pairs were enriched by peptide size exclusion chromatography (SEC) and the respective SEC fractions were analysed by LC-MS/MS. Potential cross-links were then identified by database searching including a false discovery rate of 5%. Manual validation of the high-scoring fragment spectra (see Materials and Methods) revealed 53 intra- and 13 inter-molecular cross-links in at least two out of three replicates (Supplementary Figure 2a, Supplementary Data 1). The network plot shows that the majority of these cross-links is located in the serine-rich linker region bridging the two SNARE motifs. In addition, several cross-links between the linker region

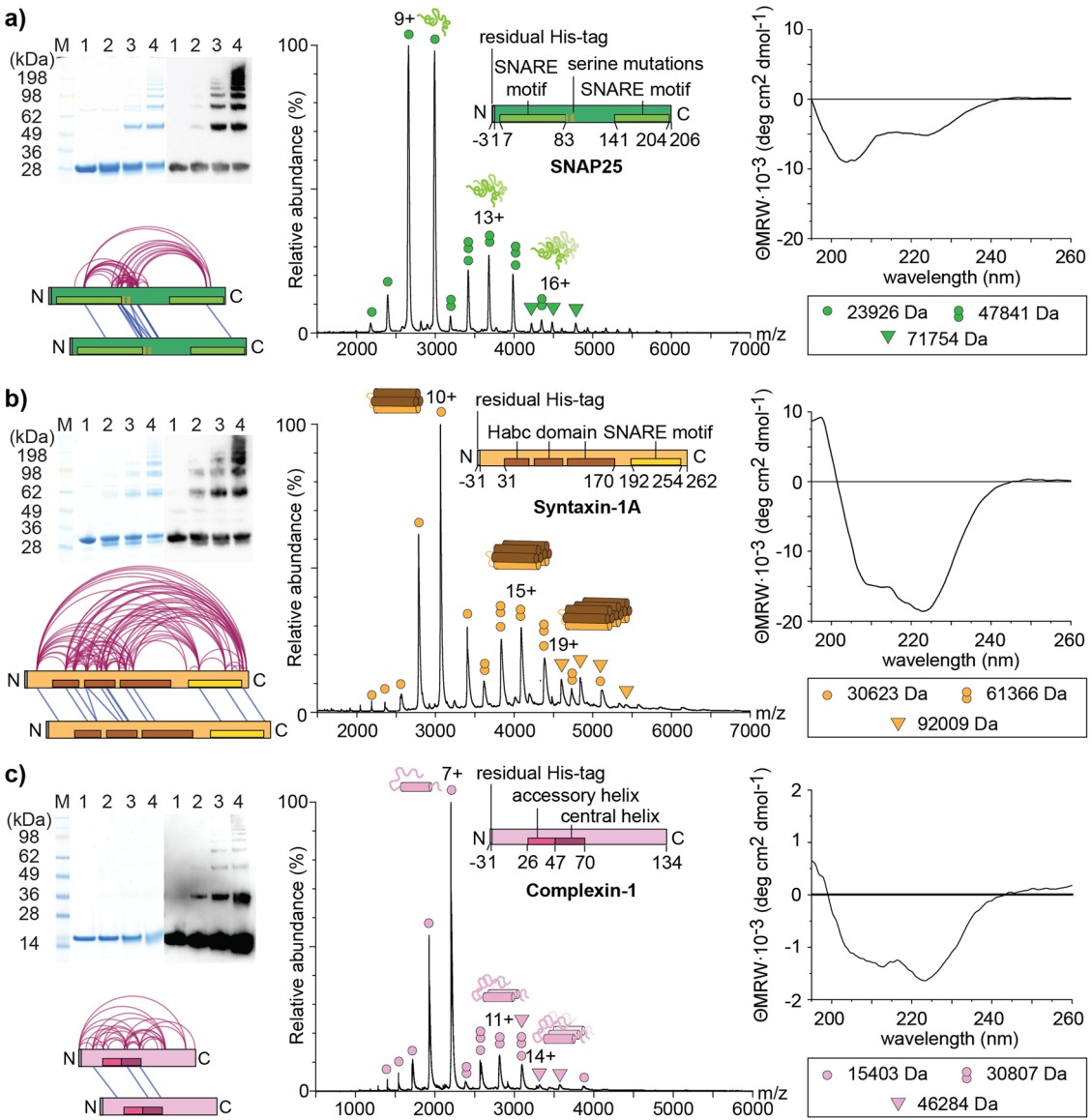

**Fig. 1 Multimerisation of SNAP25, Syntaxin-1A and Complexin-1.** Left panels: 10 µM of each protein were cross-linked with 0 mM (lane 1), 0.5 mM (lane 2), 1 mM (lane 3) and 1.5 mM (lane 4) BS3 cross-linker. Cross-linked and non-cross-linked proteins were separated by gel electrophoresis and visualised by Coomassie staining and western blot detection using anti-SNAP25, anti-Syntaxin-1A or anti-Complexin-1 antibodies, respectively. Uncropped gel and western blot images are provided in Supplementary Figure 1. For identification of cross-links, 10 µM of SNAP25 or Syntaxin-1A as well as 50 µM Complexin-1 were cross-linked with 150- or 10-fold molar excess of BS3. Cross-links identified in at least two replicates are visualized in network plots. The bars correspond to the size of the proteins. Helical structures, SNARE motifs and N-/C-termini are indicated. Middle panels: 10 µM of each protein were analysed by native MS. Charge state distributions corresponding to monomers (circle), dimers (twin-circles) and trimers (triangle) are assigned in the mass spectra. Right panels: CD spectra acquired from 30 µM protein. **a** SNAP25. **b** Syntaxin-1A. **c** Complexin-1.

and the SNARE motifs were identified suggesting high flexibility of the protein (Fig. 1a). Importantly, we also observed intermolecular cross-links which are identified by identical or overlapping peptide sequences of the cross-linked peptide pairs. These inter-molecular cross-links are mainly located in the flexible linker. Note that cross-linked peptides were enriched from a mixture of cross-linked and linear peptides by peptide SEC during sample preparation (see Materials and Methods); co-elution during reversed-phase LC and gas-phase interactions of linear and loop-linked peptides resulting in the same molecular mass as a cross-linked peptide pair[36] can therefore be neglected.

To study multimerisation of SNAP25 without covalent linkage of the protein, we performed native MS measurements. Native MS preserves non-covalent interactions of protein complexes

during their analysis in the mass spectrometer and, therefore, enables the determination of protein stoichiometries even in heterogeneous complex' mixtures[37]. Indeed, the mass spectrum of 10 µM SNAP25 shows several charge state distributions corresponding to the monomeric, dimeric and trimeric protein (Fig. 1a). To further characterise SNAP25 multimers, we analysed the secondary structure content of SNAP25 by circular dichroism (CD) spectroscopy. The CD spectrum showed two minima at 205 nm and at 225 nm characterising a mostly unstructured protein with some alpha-helical content (Fig. 1a). We conclude that SNAP25 multimers are mostly unstructured and are likely stabilised by interactions of disordered regions.

Following the same workflow, we also studied Syntaxin-1A in solution. Cross-linking with BS3, again, showed Syntaxin-1A

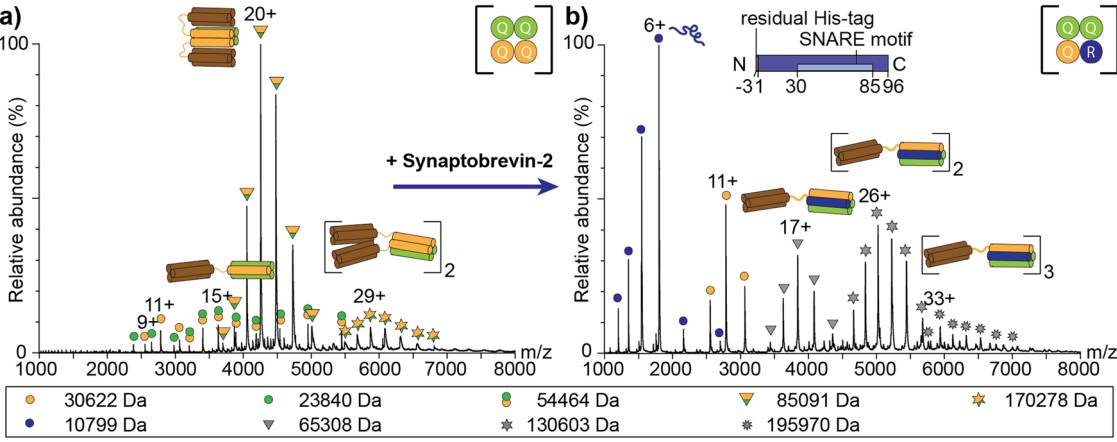

**Fig. 2 Formation and reorganization of SNAP25:Syntaxin-1A binary complexes. a** SNAP25 and Syntaxin-1A were mixed in a 1:2 ratio and analysed by native MS. Charge state series corresponding to monomeric Syntaxin-1A (orange circle) and SNAP25 (green circle) as well as binary complexes composed of SNAP25 and Syntaxin-1A in a 1:1 (green-orange twin-circles), 1:2 (orange-green triangles) and 2:4 (orange-green stars) stoichiometry were observed. **b** Addition of Synaptobrevin-2 at a 1.2 molar excess to pre-assembled binary SNAP25:Syntaxin-1A complexes results in formation of the SNAP25:Syntaxin-1A:Synaptobrevin-2 (3Q:1 R) SNARE complex. Monomeric Synaptobrevin-2 (blue circles) and Syntaxin-1A (orange circles) as well as monomeric (grey triangles), dimeric (grey hexagonal stars) and trimeric (grey nonagonal stars) SNARE complexes are assigned. The observed molecular weight is given for all proteins and complexes (see legend).

multimerisation in the absence of interaction partners; multimers up to tetramers and pentamers were observed after Coomassie staining or western blot detection (Fig. 1b, Supplementary Figure 1b). Cross-linking 10 µM Syntaxin-1A with a 150-fold molar excess of BS3 revealed 93 intra- and 13 inter-molecular interactions (Supplementary Figure 2b, Supplementary Data 2). Identified cross-links are mostly located in the three alpha-helices of the folded N-terminal Habc domain or form between the Habc domain and the C-terminal SNARE motif of Syntaxin-1A (Fig. 1b) indicating adaption of a closed conformation in the absence of other SNARE proteins as previously reported[12]. Similar to SNAP25, charge state distributions observed by native MS correspond to monomers, dimers and trimers (Fig. 1b). The CD spectrum of Syntaxin-1A showed minima at 210 nm and at 225 nm likely arising from the helical structure of the large N-terminal Habc domain (Fig. 1b).

While Syntaxin-1A due to the helical Habc domain as well as a partially structured closed conformation showed a large structural content, SNAP25 appears to be unstructured and multimers likely assemble from the flexible disordered regions of the protein suggesting stabilisation through oligomerisation. Multimer formation of both SNAREs was also reported in an early study employing site-directed spin labelling[32]. In agreement with our findings, in this study, SNAP25 multimers were found to be unfolded while Syntaxin-1A multimers showed helical arrangements.

In vivo, Complexin-1 is one of the main regulators of SNARE complex assembly[17]. We therefore also investigated Complexin-1 in solution. Using BS3 cross-linker, monomeric and dimeric Complexin-1 were observed after Coomassie staining; western blotting using a specific antibody against Complexin-1 uncovered formation of low-abundant multimers up to heptamers (Fig. 1c, Supplementary Figure 1c). For identification of specific protein interactions, 50 µM Complexin-1 were cross-linked with a 10-fold molar excess of BS3 (Fig. 1c). 31 intra- and 3 inter-molecular cross-links were identified in this experiment (Supplementary Figure 2c, Supplementary Data 3). Protein interactions of Complexin-1 were mainly captured between the N-terminal domain and the accessory helix as well as the C-terminal domain and the central helix in agreement with a high flexibility of the protein's termini as suggested previously[22].

In agreement with these cross-linking experiments, the native mass spectrum of Complexin-1 reveals three charge state distributions corresponding to monomers, dimers and trimers (Fig. 1c). The CD spectrum of Complexin-1 revealed a helical structure as confirmed by two minima at 210 nm and 225 nm (Fig. 1c). Multimerisation of Complexin-1 suggests that stabilisation of the protein in the absence of interaction partners occurs in a similar manner as observed for the individual SNAREs.

**Synaptobrevin-2 rearranges the SNAP25:Syntaxin-1A binary complex.** We then moved forward and explored the formation of binary SNARE complexes. We first targeted SNAP25 and Syntaxin-1A, which have previously been proposed to form a 1:1 acceptor complex for Synaptobrevin-2 preceding the formation of the full SNARE complex[31]. For this, the two proteins were mixed in a 1:1 ratio and assembled complexes were analysed by native MS. The mass spectrum showed several charge state distributions corresponding to the individual proteins as well as SNAP25:Syntaxin-1A complexes with a 1:1, 2:1, 1:2, 2:2 and 2:4 stoichiometry (Supplementary Figure 3). The predominant species corresponds to the 1:2 SNAP25:Syntaxin-1A complex suggesting the formation of a 4Q:0R four-helix bundle, in which the Synaptobrevin-2 binding site is occupied by an additional helix of Syntaxin-1A[4]. Formation of this complex in a liposome fusion system was previously described as kinetically trapped[30].

To specifically assess formation of this dead-end acceptor complex, we shifted the equilibrium of complex formation and mixed SNAP25 and Syntaxin-1A in a 1:2 molar ratio. Indeed, the complex with the highest intensity of the acquired mass spectrum is the 1:2 complex suggesting that this stoichiometry is preferred and possibly most stable in the absence of Synaptobrevin-2 (Fig. 2a). Low-intense SNAP25:Syntaxin-1A complexes with 1:1 and 2:4 stoichiometry were also observed suggesting that complex formation is, at least to some degree, variable and dynamic. While the 2:4 complex is a dimer of the 1:2 species, the 1:1 complex represents the expected active acceptor complex providing a binding site for Synaptobrevin-2 when available.

We next questioned whether the observed 4Q:0R SNAP25:Syntaxin-1A complex indeed resembles the ternary SNARE complex and, to this end, performed CD spectroscopy

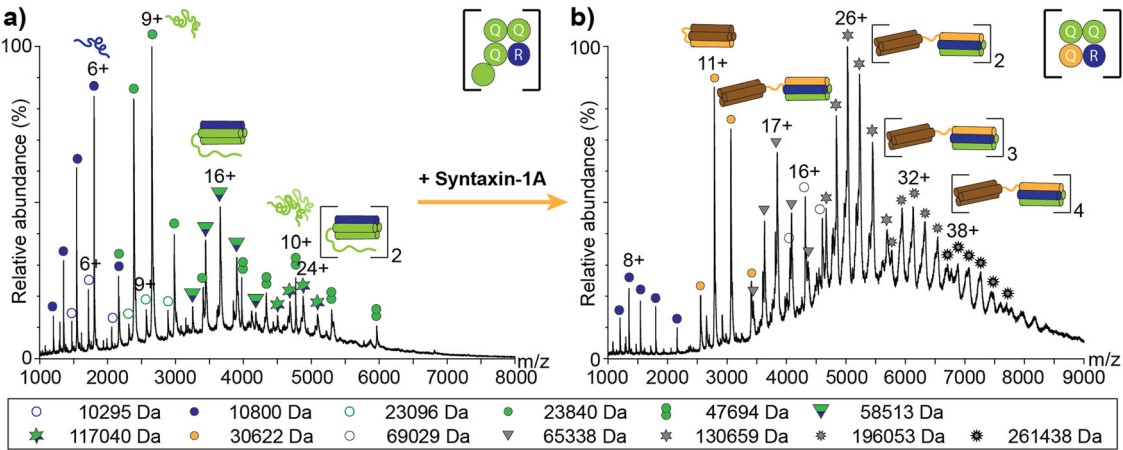

**Fig. 3 Formation and rearrangement of the SNAP25:Synaptobrevin-2 binary complex. a** SNAP25 and Synaptobrevin-2 were mixed in a 1:1 ratio and analysed by native MS. Charge state series corresponding to monomeric Synaptobrevin-2 (blue circle) and SNAP25 (green circle), dimeric SNAP25 (green twin-circles) as well as binary complexes composed of SNAP25 and Synaptobrevin-2 with a stoichiometry of 2:1 (green-blue triangle) and 4:2 (green-blue hexagonal stars) were observed. **b** Upon equimolar addition of Syntaxin-1A, monomeric (grey triangles), dimeric (grey hexagonal stars), trimeric (grey nonagonal stars) and tetrameric (grey dodecagonal) SNARE complexes were observed. The observed molecular weight is given for all proteins and complexes (see legend).

of the binary protein mixture (Supplementary Figure 4a). The acquired CD spectra of the individual proteins revealed random coil structures with some helical content for SNAP25 and helical structures of Syntaxin-1A (see above). After mixing of the two proteins, a meaningful change in the helical content when compared with the theoretically calculated CD spectrum, which would be determined when the two proteins do not interact, was not observed (Supplementary Figure 4a). However, the arrangement of the helices in the closed conformation of Syntaxin-1A and the potentially formed SNAP25:Syntaxin-1A complex cannot be distinguished and structural rearrangements might have occurred.

To interrogate whether the 1:2 SNAP25:Syntaxin-1A complex is kinetically trapped as suggested for the membrane-bound complex, we added a 1.2 molar excess of Synaptobrevin-2, the naturally preferred interaction partner of the acceptor complex, and followed complex formation by native MS (Fig. 2b). Surprisingly, the acquired mass spectrum showed none of the above identified SNAP25:Syntaxin-1A complexes; instead, monomeric, dimeric and trimeric fully assembled (i.e., 3Q:1R) SNARE complexes as well as monomeric Synaptobrevin-2 and Syntaxin-1A were immediately observed. These experiments, therefore, confirm a higher affinity of Synaptobrevin-2 for the R-SNARE binding site of the acceptor complex when compared with the Q-SNARE Syntaxin-1A. The absence of monomeric SNAP25 suggests rearrangement of the 1:2 SNAP25:Syntaxin-1A complex rather than its complete disassembly and reassembly; although, an equilibrium between the 1:1 and 1:2 complexes would also explain formation of the ternary SNARE complex without dissociation of monomeric SNAP25. Notably, the mass spectra of both, the dead end complex (Fig. 2a) and the fully assembled SNARE complex (Fig. 2b), reveal multimerisation of the two complexes.

**Formation of binary complexes imitating SNARE complex stoichiometry.** We next studied complex formation between SNAP25 and Synaptobrevin-2. Formation of a complex between SNAP25 and Synaptobrevin-2 was controversially discussed in several previous studies. While the SNAP25:Synaptobrevin-2 complex was suggested to represent the first step of membrane fusion in PC12 cells[38], other studies did not observe stable complexes[39] or only week interactions[40,41]. We, therefore, mixed the two SNAREs in a 1:1 molar ratio and explored formation of

complexes by native MS. The mass spectrum showed charge state distributions corresponding to monomeric Synaptobrevin-2 and SNAP25 as well as low abundant dimeric SNAP25 (Fig. 3a). Importantly, two additional charge state series corresponding to binary complexes containing SNAP25 and Synaptobrevin-2 at a stoichiometry of 2:1 and a dimer thereof with a stoichiometry of 4:2 were also observed. Considering a four-helical assembly similar to the ternary SNARE complex, Synaptobrevin-2 contributes one alpha-helix and the two SNAP25 molecules contribute three alpha-helices to the complex supposing that one of the four SNARE motifs is located exterior of the four-helix bundle. Exclusion of one SNAP25 SNARE motif is in agreement with a 3Q:1R SNARE four-helix bundle. Note that the SNAP25 dimer showed lower charge states than the 2:1 SNAP25:Synaptobrevin-2 complex. In some cases, labile complexes dissociate in the gas-phase resulting in a highly charged dissociated monomer and a so-called stripped complex carrying less charges; accordingly, we suppose that the SNAP25 dimer forms during gas-phase dissociation of the 2:1 complex rather than in solution.

Again, CD spectroscopy was used to evaluate complex formation between the two proteins. While the individual proteins were found to be unstructured, comparing the CD spectrum of the protein mixture with the calculated CD spectrum revealed a signal shift from 203 nm towards 208 nm as well as an increase in ellipticity at 220 nm indicating a decrease in the amount of disordered regions and formation of alpha-helices (Supplementary Figure 4b). The presence of helical structures suggests that a helix bundle similar to the ternary SNARE complex was indeed formed. Whether the additional SNARE motif is structured or unstructured remains, however, elusive.

To challenge the observed stoichiometry of the binary SNAP25:Synaptobrevin-2 complex, we varied the incubation ratios of the two proteins. However, when incubating SNAP25 and Synaptobrevin-2 in 1:2 and 2:1 mixing ratios, the same complex with a stoichiometry of 2:1 formed (Supplementary Figure 5) confirming that the assembly of a four-helical 3Q:1R binary complex is indeed preferred.

To test whether the 2:1 SNAP25:Synaptobrevin-2 complex can rearrange into the fully assembled SNARE complex, we added equimolar amounts of Syntaxin-1A and analysed the formed complexes. Indeed, Syntaxin-1A replaces one of the SNAP25 alpha-helices. The observed complexes, now, present the fully

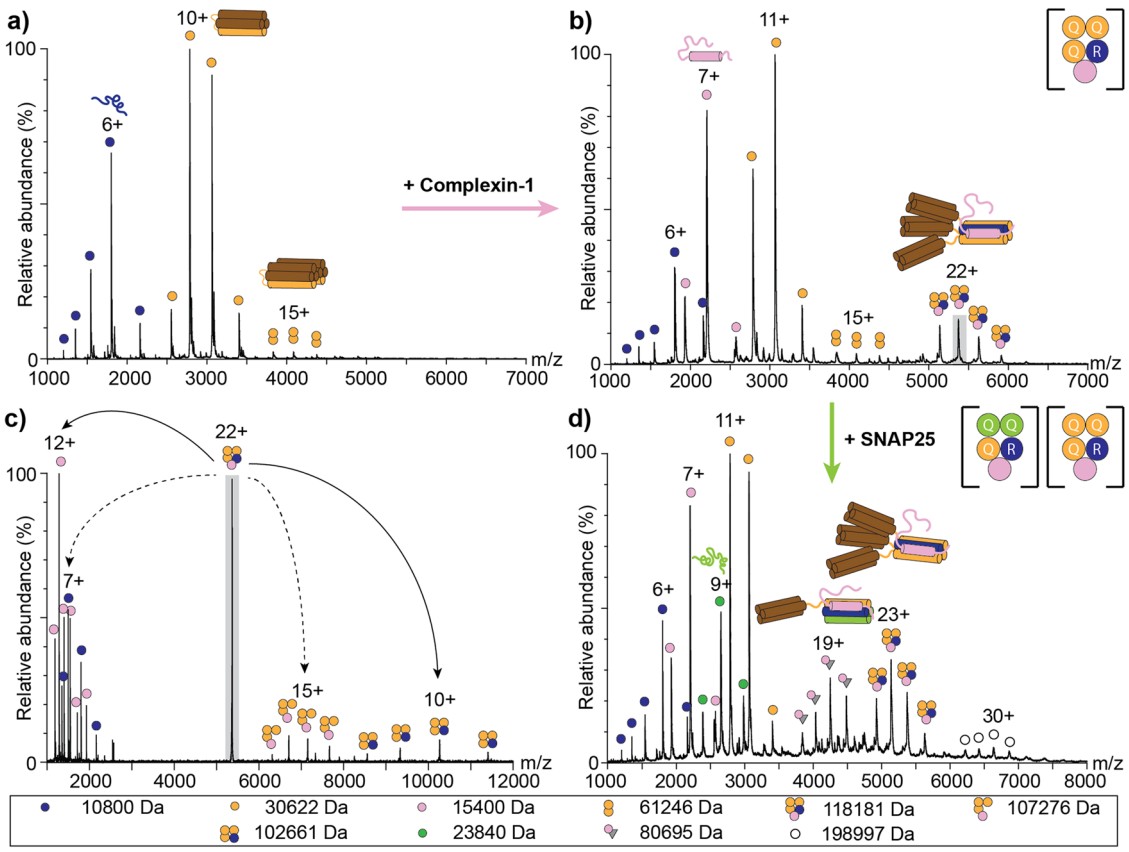

**Fig. 4 The binary Synaptobrevin-2:Syntaxin-1A complex is stabilised by Complexin-1. a** Synaptobrevin-2 and Syntaxin-1A were mixed in a 1:1 ratio and analysed by native MS. Charge state series corresponding to monomeric Synaptobrevin-2 (blue circle) and Syntaxin-1A (orange circles) were observed. **b** Addition of Complexin-1 to Synaptobrevin-2 and Syntaxin-1A results in formation of the 1:3:1 Synaptobrevin-2:Syntaxin-1A:Complexin-1 complex. Charge state series of monomeric Synatobrevin-2 (blue circles), Syntaxin-1A (orange circles) and Complexin-1 (pink circles), dimeric Syntaxin-1A (orange twin-circles) as well as the 1:3:1 complex (orange-blue-pink circles) are assigned. **c** The 22+ charge state of the 1:3:1 complex (highlighted in grey in panel **b** was selected for collision induced dissociation. Dissociated peripheral subunits (Synaptobrevin-2, blue circles; Complexin-1, pink circles) as well as two stripped complexes (3:1 Syntaxin-1A:Complexin-1, orange-pink circles; 3:1 Syntaxin-1A:Synaptobrevin-2, orange-blue circles) are assigned. **d** Complex formation upon equimolar addition of SNAP25 to the pre-assembled 1:3:1 Synaptobrevin-2:Syntaxin-1A:Complexin-1 complex was followed by native MS. Monomeric Synaptobrevin-2 (blue circles), SNAP25 (green circles) and Complexin-1 (pink circles) as well as the observed complexes (grey triangle-pink circle; blue-yellow-pink circles) are assigned. The observed molecular weight is given for all proteins and complexes (see legend).

assembled (3Q:1R) ternary SNARE complex and oligomers thereof (Fig. 3b).

**Complexin-1 stabilises interactions of the binary Synaptobrevin-2:Syntaxin-1A complex.** Next, we also investigated complex formation between Synaptobrevin-2 and Syntaxin-1A. The native mass spectrum of a 1:1 mixture shows two charge state distributions with high intensities corresponding to the monomers of Synaptobrevin-2 and Syntaxin-1A; additional charge state series were not observed suggesting that formation of a complex between Synaptobrevin-2 and Syntaxin-1A is impeded (Fig. 4a). However, oligomerisation of the monomers as observed above for the individual proteins was also prevented suggesting that the two SNAREs transiently interact without forming a stable binary complex that can be captured by native MS. We therefore analysed the protein mixture by CD spectroscopy and found that, the alpha-helical content decreased compared to the theoretical CD spectrum (Supplementary Figure 4c) revealing structural rearrangements, which likely corresponds to dissociation of the SNARE motif of Syntaxin-1A from its Habc domain, thus, being available to interact with the SNARE motif of Synaptobrevin-2 to form a complex. The decreased helicity in the CD spectrum and the absence of a stable complex in the native mass spectrum

indicates the necessity for SNAP25 or additional proteins to stabilise interactions between Synaptobrevin-2 and Syntaxin-1A.

Complexin-1 is a well-described regulator of SNARE complex-mediated membrane fusion[17]. Interestingly, two high-resolution crystal structures of the fully assembled complex[23,42] and an early site-directed mutagenesis study[22] reveal binding of Complexin-1 at the interface of Syntaxin-1A and Synaptobrevin-2. Therefore, we speculate that Complexin-1 might be able to stabilise the transient Synaptobrevin-2:Syntaxin-1A interface. Based on this assumption, we added equimolar amounts of Complexin-1 to the pre-incubated proteins. Indeed, the obtained mass spectrum revealed an additional charge state series corresponding to a ternary Synaptobrevin-2:Syntaxin-1A:Complexin-1 complex with a stoichiometry of 1:3:1 (Fig. 4b). We assume that the two SNAREs resemble the four helical bundle of the trimeric SNARE complex by substituting the two alpha-helices of SNAP25 for two additional alpha-helices of Syntaxin-1A (3Q:1R) and that Complexin-1 does not replace or supplement SNARE helices but rather associates with the complex at the periphery.

To verify the peripheral position of Complexin-1, we selected the 22+ charge state of the complex for collision-induced dissociation. Increasing the collisional energy during native MS causes asymmetric dissociation of an exposed protein subunit that requires the least activation energy for unfolding, resulting in

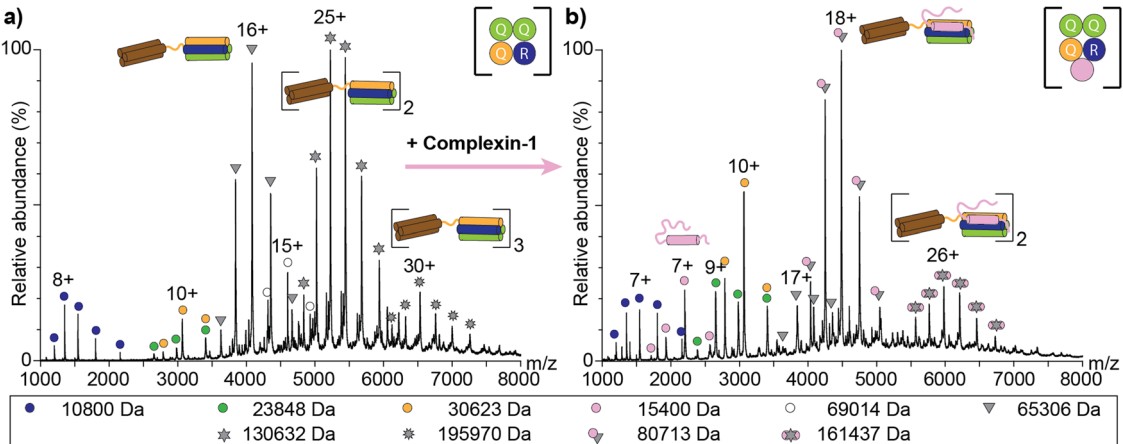

**Fig. 5 Multimerisation of the fully assembled SNARE complex is prevented by Complexin-1. a** Synaptobrevin-2, SNAP25 and Syntaxin-1A were mixed in a 1:1:1 ratio and analysed by native MS. Charge state series corresponding to the monomeric proteins (Synaptobrevin-2, blue; Syntaxin-1A, orange; SNAP25, green) as well as monomeric (grey triangles), dimeric (grey hexagonal stars) and trimeric (grey nonagonal stars) SNARE complexes were observed. **b** Upon addition of Complexin-1, SNARE complex oligomerisation is reduced. Charge states corresponding to the SNARE:Complexin-1 (grey triangle-pink circle) as well as a dimeric SNARE:Complexin-1 (grey hexagonal star-pink circle) complex were observed. The observed molecular weight is given for all proteins and complexes (see legend).

a highly charged monomer and a stripped complex composed of the remaining protein subunits[43,44]. Here, dissociation of Synaptobrevin-2 and Complexin-1 yielded two stripped complexes, namely the 3:1 Syntaxin-1A:Complexin-1 complex and the 3:1 Syntaxin-1A:Synaptobrevin-2 complex (Fig. 4c). Dissociation of Complexin-1 confirms the peripheral position of the protein at the surface of the complex. The second dissociation event expelling Synaptobrevin-2 was surprising, however, indicates least stable incorporation of the protein into the helix bundle.

As Complexin-1 stabilises the binary Synaptobrevin-2:Syntaxin-1A complex through the binding interface, we next interrogated whether the two additional Syntaxin-1A molecules can be replaced by SNAP25 in the pre-assembled complex. For this, we first incubated Synaptobrevin-2, Syntaxin-1A and Complexin-1 forming the complex with a stoichiometry of 1:3:1 described above. We then added an equimolar amount of SNAP25 and followed complex rearrangement over time (Supplementary Figure 6). Immediately after addition of SNAP25, only the pre-assembled complex was observed. However, when incubating the proteins for several minutes, formation of the fully assembled SNARE complex binding one Complexin-1 molecule was monitored (Fig. 4d and Supplementary Figure 6). After longer incubation times above 10 minutes, this complex was the sole species observed. We conclude that Complexin-1 indeed stabilises the Synaptobrevin-2:Syntaxin-1A interface while other molecules of the helix bundle can be replaced. The observation that SNAP25 incorporates over a time period lasting several minutes suggests that blocking one interface of the four-helical assembly decelerates complex rearrangements. Alternatively, disassembly of the SNARE intermediate might be hampered, therefore, decelerating formation of the SNARE:Complexin-1 complex.

To test whether Complexin-1 also stabilises other interfaces of binary SNARE complexes, we pre-assembled binary SNAP25:Syntaxin-1A complexes (see above) and incubated these complexes with Complexin-1. The acquired mass spectrum revealed one additional charge state series corresponding to monomeric Complexin-1 (Supplementary Figure 7a). Charge states of a complex including Complexin-1 were not observed. We then also pre-assembled the third binary complex composed of SNAP25:Synaptobrevin-2 at a stoichiometry of 2:1 (see

above) followed by addition of Complexin-1. Again, the mass spectrum revealed a charge state series corresponding to monomeric Complexin-1; additional charge state series of additional complexes were not observed (Supplementary Figure 7b). We reason that Complexin-1 specifically binds the Synaptobrevin-2:Syntaxin-1A interface. Protein interfaces containing either Synaptobrevin-2 or Syntaxin-1A do not stably associate with Complexin-1. To further prove this assumption, we incubated the three SNAREs with Complexin-1 in independent experiments. All three mass spectra show charge states of the individual proteins as well as Complexin-1; binding of Complexin-1 to individual SNAREs was, however, not observed (Supplementary Figure 8a-c).

**Complexin-1 prevents multimerisation of the fully assembled SNARE complex.** Finally, we explored the assembly of the ternary 3Q:1R SNARE complex. For this, the three SNARE proteins were mixed in a 1:1:1 ratio and incubated overnight. Assembled complexes were then identified by native MS. The acquired mass spectrum showed charge state series corresponding in mass to one, two and three SNARE complexes (Fig. 5a) revealing multimerisation of the fully-assembled complex as also observed above for binary SNARE complexes (see above). Importantly, the monomeric and dimeric SNARE complexes showed comparable intensities suggesting stable multimer formation in solution. We next selected the 16+ and 25+ charge states of the monomeric and dimeric SNARE complexes for collision-induced dissociation of peripheral or weakly-associated protein subunits; Synaptobrevin-2 readily dissociated in these measurements (Supplementary Figure 9a) indicating least stable incorporation of the smallest subunit into the four-helix bundle in contrast to the acceptor complex formed by SNAP25 and Syntaxin-1A.

The stimulating and inhibitory roles of Complexin-1 during SNARE-mediated membrane fusion are still controversially discussed[27,28]. We found that Complexin-1 specifically stabilises the Synaptobrevin-2:Syntaxin-1A interface and, therefore, also investigated interactions of Complexin-1 with the fully assembled SNARE complex. For this, an equimolar amount of Complexin-1 was added to the pre-assembled SNARE complexes and interactions between the SNARE complex and Complexin-1 were

followed by native MS. The acquired mass spectrum shows two charge state distributions corresponding to the SNARE:Complexin-1 complex as well as a dimer of this complex (Fig. 5b). Note that the intensity of the dimeric SNARE:Complexin-1 complex is remarkably reduced when compared with the dimeric SNARE complex observed in the absence of Complexin-1. We conclude that Complexin-1 stoichiometrically binds the SNARE complex and, importantly, reduces its self-association as demonstrated by the absence of higher SNARE multimers.

Collision induced dissociation of the monomeric and dimeric SNARE:Complexin-1 complex further revealed dissociation of Complexin-1 and Synaptobrevin-2 (Supplementary Figure 9b), again confirming the peripheral position of Complexin-1 and least stable incorporation of Synaptobrevin-2 into the SNARE helix bundle. We assume that the Synaptobrevin-2:Syntaxin-1A:Complexin-1 complex as well as the SNARE:Complexin-1 complex assemble in a similar manner.

**Protein interactions in the SNARE:Complexin-1 complex**. We next studied interactions formed between the ternary SNARE complex and Complexin-1 in detail. Available high resolution structures of the SNARE complex provide insights into the structural arrangements within the SNARE four-helix bundle[5,45]. Binding of Complexin-1 to the SNARE complex was found to be realised antiparallel through interactions of the central helix with a groove formed by Synaptobrevin-2 and Syntaxin-1A on the surface of the SNARE helix bundle[23,42,46]. To shed light on these interactions and specifically include flexible regions of the proteins missing in the available high-resolution structures, we incubated the assembled SNARE:Complexin-1 complex with BS3 cross-linker introducing covalent linkages between reactive groups in spatial proximity. The cross-linked complex was then denatured using urea followed by digestion with trypsin and enrichment of cross-linked peptide pairs (see Methods for details). After LC-MS/MS and database searching including a false-discovery rate of 5%, 337 cross-links including 154 intra- and 162 inter-molecular interactions as well as 21 cross-links corresponding to homo-oligomers were identified in the SNARE:Complexin-1 complex. Of these, 100 intra- and 82 inter-molecular cross-links were identified in at least two of three replicates confirming a good reproducibility of the cross-linking experiments (Supplementary Figure 10, Supplementary Data 4).

Inter-molecular cross-links were first visualised in a network plot (Supplementary Figure 11). Interactions between the three SNARE proteins mainly form between the SNARE motif of Synaptobrevin-2, the flexible linker of SNAP25 as well as the C-terminus of Syntaxin-1A. In addition, many cross-links between the Habc domain of Syntaxin-1A and the other two SNAREs are observed indicating the high flexibility of this domain in the assembled complex (Supplementary Figure 11). Interactions between Complexin-1 and the SNAREs confirm antiparallel binding of the protein. The network plot further reveals interactions of Complexin-1 with the C-termini of Synaptobrevin-2 and Syntaxin-1A, the flexible linker of SNAP25 as well as the Habc domain of Syntaxin-1A (Supplementary Figure 11).

For three-dimensional visualisation and validation, identified cross-links were mapped onto available high-resolution structures and predicted structural models. For this, we first visualised cross-links observed between the three SNAREs (Supplementary Figure 12). Of these, 8 inter-molecular cross-links between Synaptobrevin-2, SNAP25 and Syntaxin-1A were included in the high-resolution structure of the four-helix bundle (Supplementary Figure 12). Additional cross-links correspond to residues which are located in flexible linkers or termini and are, therefore,

not present in the crystal structure. Using an AlphaFold2 prediction of the SNARE complex enabled visualisation of all 16 cross-links including cross-links formed with disordered regions of the SNAREs such as the flexible linker of SNAP25. In many cases, identified cross-links correlate well with the high-resolution structure and cross-linking distances are in the expected range below 30 Å. However, in some cases, the expected cross-linking distance is exceeded (Supplementary Figure 12a and b). This might be due to multimerisation of the SNARE complex, which was observed in the absence and, to a lower extent, in the presence of Complexin-1 (see above). Indeed, when a second high-resolution structure of the SNARE complex is included in this analysis, all cross-linking distances are in the expected range (Supplementary Figure 12c). Note that this arrangement requires antiparallel orientation of the two SNARE complexes to satisfy the observed cross-links. We, therefore, verified the antiparallel arrangement of the SNARE complex in an independent approach using ColabFold[47], which combines homology searching with AlphaFold2 or RoseTTAFold for predictions of protein structures and complexes[48,49]. For this, the protein sequences were submitted to the ColabFold webserver enabling visualisation of flexible regions and dimerization of the proteins. Importantly, the predicted dimeric SNARE complex also aligns in an antiparallel manner confirming our assumption (Supplementary Figure 12d).

Our final goal was to build a model showing interactions between Complexin-1 and the SNARE complex. As a high-resolution structure of full-length Complexin-1 is not available, we first generated a model of the full-length protein using AlphaFold[48]. This predicted structure contained the central alpha helix which was previously resolved[23] as well as unstructured N- and C-termini with some helical elements. For model preparation, we first determined the position of the central helix of Complexin-1 with respect to the SNARE complex by aligning a high-resolution structure of the complete SNARE complex with a structure of the SNARE complex containing the central helix of Complexin-1 but lacking parts of the structures for Synaptobrevin-2[23]. We then aligned the full-length model of Complexin-1 with the central helix of this temporary model resulting in a model comprising the complete SNARE complex as well as the generated full-length structure of Complexin-1. An available high-resolution structure of the Habc domain was then also added to the arrangement[50]. Missing structural elements such as the flexible linker of SNAP25, the C-terminus of Syntaxin-1A and the flexible linker of Syntaxin-1A connecting the SNARE motif and the Habc domain were indicated manually. Using this assembly, we then highlighted identified cross-links in the structures and added missing residues in the flexible loops manually. The N-terminal unstructured loop of Complexin-1 was repositioned manually to justify all interactions (Fig. 6).

Again, we compared our manually assembled model with an unbiased prediction using ColabFold, and visualised observed cross-links in this structural model. The obtained model greatly resembles the manually assembled model (Supplementary Figure 13). Accordingly, the central helix of Complexin-1 occupies the Synaptobrevin-2:Syntaxin-1A interface in an antiparallel orientation and the Habc domain of Syntaxin-1A is directed away from the helix bundle. However, the flexible N- and C-terminal regions of Complexin-1 are unstructured and randomly oriented in this prediction; an engagement in the complex was not achieved. We assume that, due to missing high-resolution templates, the structure prediction of disordered regions of Complexin-1 failed. These differences are expressed in many over-length cross-links when plotting the experimentally observed cross-links into the predicted complex (Supplementary Figure 13). Nonetheless, the core complex of the structural prediction and the manually assembled model agree well and,

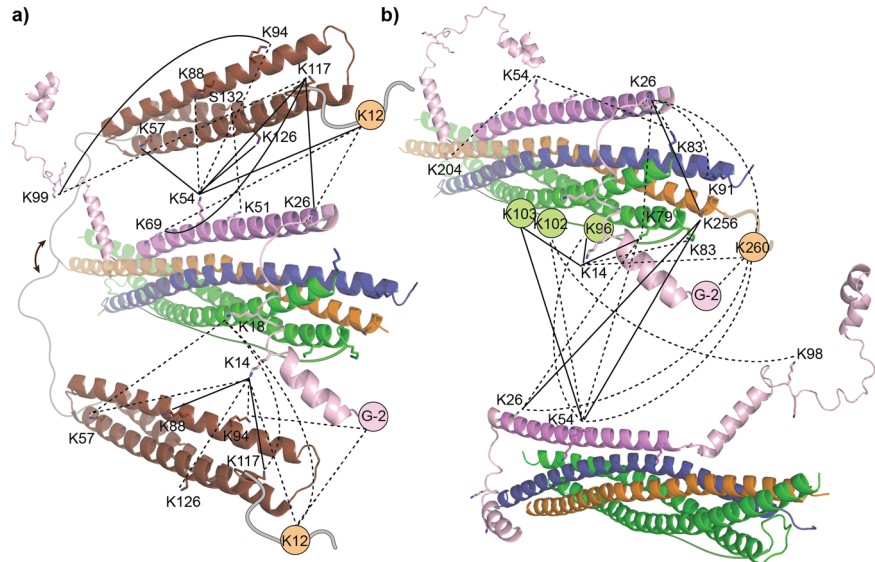

**Fig. 6 Model of the SNARE:Complexin-1 complex.** Complexin-1 (pink), Synaptobrevin-2 (blue), the SNARE alpha-helix of Syntaxin-1A (orange), the Habc domain of Syntaxin-1A (brown) and SNAP25 (green) are shown. Missing loops were added manually. Cross-linked lysine residues are labelled and shown in stick representation or are indicated in circles. Cross-links are visualised as dotted (identified in two replicates) and solid (identified in three replicates) lines. **a** Interactions between the Habc domain of Syntaxin-1A and Complexin-1. Interactions are visualised by showing two copies of the Habc domain. Flexibility of the linker is indicated (arrow). **b** Interactions between Complexin-1 and the SNARE complex. Two copies of the SNARE complex are shown.

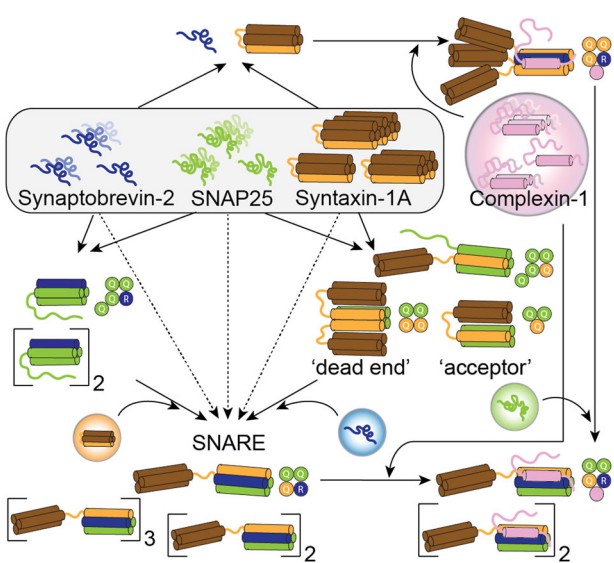

**Fig. 7 Roadmap of SNARE complex assembly.** Complexes observed by native MS when mixing Synaptobrevin-2 (blue), SNAP25 (green), Syntaxin-1A (orange) and Complexin-1 (pink) are shown as cartoons. The stoichiometry of the complexes as observed in native mass spectra is indicated.

importantly, experimentally observed cross-links allow refinement of the flexible disordered regions in a manual assessment.

Inspecting the two models, we make the following conclusions: First, the linker connecting the SNARE motif and the Habc domain of Syntaxin-1A is indeed highly flexible resulting in interactions of the Habc domain with the central helix of Complexin-1 (indicated by cross-links with residues K26, K51, K54 and K69) as well as the N-terminal unstructured loop of Complexin-1 (including residues G-2, K14 and K18) (Fig. 6a). Second, the N-terminal unstructured loop of Complexin-1 overlaps the C-termini of the four SNARE motifs including

interactions with Synaptobrevin-2 (K83 and K91), Syntaxin-1A (K256 and K260) and SNAP25 (K79 and K83). Third, cross-links between Complexin-1 (K26, K54 and K98), SNAP25 (K79, K96, K102 and K103) and Syntaxin-1A (K 256 and K260) are in agreement with interactions between two SNARE complexes (Fig. 6b).

The cross-links of Complexin-1 in our model all satisfy the distance restraints and, importantly, suggest a mechanism preventing SNARE multimerisation upon Complexin-1 binding. Accordingly, Complexin-1 is clamping the C-termini of the SNAREs in the four-helix bundle and, therefore, occupying the binding interface of a second SNARE complex. In agreement with this, we observed mostly monomeric and to a lower extent dimeric SNARE:Complexin-1 complexes in our native mass spectra (Fig. 5b). We conclude that Complexin-1 sterically hinders multimerisation of SNARE complexes.

## Discussion
Using native MS, we compiled a roadmap for SNARE complex assembly (Fig. 7) and uncovered preferred sub-complexes and their stoichiometry. Integrating Complexin-1 into the assembly roadmap provides clues on its role during SNARE complex assembly and membrane fusion. In summary, we made the following observations: (i) SNAP25, Syntaxin-1A and Complexin-1 multimerise in the absence of other SNARE proteins. (ii) Binary SNARE complexes preferably resemble the native 3Q:1R SNARE assembly; in the absence of the R-SNARE Synaptobrevin-2, 4Q:0R complexes are formed. The preferred stoichiometry of these complexes could not be challenged by varying the protein mixing ratios. (iii) The SNAP25:Synaptobrevin-2 complex unexpectedly included an additional copy of SNAP25; we assume that only one SNARE motif of the additional SNAP25 copy is integrated into this binary complex. (iv) The binary Synaptobrevin-2:Syntaxin-1A complex could not be captured by native MS; this complex was, however, stabilised upon binding of Complexin-1. Importantly, Complexin-1 does not replace one of the SNAREs in this assembly and rather binds the 3Q:1R complex. Interestingly, Complexin-1 does not bind individual SNARE proteins or binary

SNARE complexes that contained either Synaptobrevin-2 or Syntaxin-1A. (v) Multimerisation of the ternary SNARE complex was remarkably reduced in the presence of Complexin-1. (vi) Our model of the SNARE:Complexin-1 complex reveals arrangement of the flexible regions of Complexin-1 when bound to the SNARE complex. This model suggests that multimerisation of the SNARE complex is impeded upon Complexin-1 binding due to sterical hindrance.

Multimerisation of SNARE proteins was previously reported for recombinantly expressed soluble constructs, transmembrane domains and full-length proteins[32,33,51,52] as well as for SNAREs isolated from their native membranes[53,54]. In this study, we found that Syntaxin-1A, SNAP25 and Complexin-1, similar to Synaptobrevin-2[33], multimerise (Fig. 7); a defined oligomeric state was, however, for none of the proteins identified. Differences in the oligomeric state observed by gel electrophoresis/western blotting or native MS are likely attributed to the different sensitivity of the methods. Notably, multimers appear to be largely unstructured with some helical content and interactions were mostly identified in flexible regions of the proteins. We, therefore, assume that multimerisation stabilizes the unstructured proteins without formation of a structured protein assembly. Our assumption is supported by the absence of oligomers when interaction partners were added and SNARE (sub-) complexes formed.

Binary complexes, on the contrary, formed defined complexes that imitated the 3Q:1R SNARE assembly, provided R- and Q-SNAREs were available. While a single SNARE appears to be unable to form a complex that is similar to the functionally active, ternary complex; the addition of a second SNARE allows formation of stable complexes. In all binary complexes, integration of more than one R-SNARE (Synaptobrevin-2) was avoided; formation of the unexpected Synaptobrevin-2:SNAP25 complex, which likely integrated an additional alpha-helix of a second SNAP25 molecule, suggests that two arginine residues in the central ionic layer destabilize the SNARE four-helix bundle. Therefore, in the absence of Synaptobrevin-2, 4Q:0R complexes were formed. Our observations on the stoichiometry of binary complexes is in agreement with a previous study examining possible combinations of arginine and glutamine residues in the central layer of the SNARE four-helix bundle[55]. This study revealed that two arginine residues in the central layer are highly disruptive due to electrostatic and steric hindrances. Four glutamine residues, on the other hand, did not affect SNARE formation, although these complexes were less stable[55].

For all binary SNARE complexes, we found that the missing third SNARE component readily incorporated, albeit at different time-scales. We conclude that the ternary, fully assembled SNARE complex is the preferred and presumably the most stable complex. This is particularly true for the 2:1 Syntaxin-1A:SNAP25 complex which was previously described as kinetically trapped[30]. While this complex was assembled from membrane-anchored protein variants, we assume that soluble protein variants used here allow disassembly of binary complexes and reassembly of the ternary complex. An equilibrium between the 2:1 dead end acceptor complex and the 1:1 active acceptor complex, which were both observed in our mass spectra, might accelerate complex' rearrangements or disassembly-reassembly processes. Similar assumptions were made previously when studying the mesoscale organisation of the plasma membrane in PC12 cells[56]. This study revealed co-existing tight clusters of alpha-helical Syntaxin-1A bundles and diffuse clusters of SNAP25, both interacting only at their peripheries. Diffuse SNAP25 clusters are recruited to Syntaxin-1A clusters through alpha-helical interactions of the N-terminal SNARE motif of SNAP25 showing Syntaxin-1A:SNAP25 complexes only at the interfaces of the clusters.

Higher abundance of SNAP25 in the membranes, as observed in PC12 cells, prevents formation of the dead end acceptor complex and drives formation of 1:1 active acceptor complexes[56]. However, the abundance of SNAP25 differs between species and additional regulators such as Munc18 and Munc13 regulate formation of the acceptor complex[12–14].

Note that incorporation of SNAP25 into the 1:3:1 Synaptobrevin-2:Syntaxin-1A:Complexin-1 complex was comparatively slower than into other pre-assembled complexes. One explanation is stable helix formation of Syntaxin-1A oligomers[32,57]. Accordingly, an early study proposed trimer formation of Syntaxin-1A[51]; trimeric Syntaxin-1A appears to be comparably stable and subunit replacement in this complex is therefore slower. Furthermore, the Synaptobrevin-2:Syntaxin-1A interface in this complex is occupied by Complexin-1. We assume that antiparallel binding of its central helix locks the interface between the two subunits and, assuming a similar arrangement in the 1:3:1 Synaptobrevin-2:Syntaxin-1A:Complexin-1 complex as in our model of the SNARE:Complexin-1 complex (Fig. 5), Complexin-1 clamps the C-termini of the proteins; the disassembly-reassembly processes might be hampered or even impeded, the latter requiring replacement of two Syntaxin-1A molecules by one SNAP25 molecule.

In a similar manner, the complete incorporation of Syntaxin-1A into the pre-assembled 2:1 SNAP25:Synaptobrevin-2 complex required overnight incubation. Extended time scales for re-arrangement of this binary complex might be caused by the additional SNARE motif or by antiparallel arrangements of the proteins. Assuming a four-helical bundle similar to the fully assembled ternary SNARE complex, the additional fifth SNARE motif is likely unfolded and not involved in the formation of the complex. This unstructured loop might interact with the binding interfaces of the complex or sterically hinder rearrangements. The question whether the N- or the C-terminal SNARE motif of SNAP25 is engaged in the helical arrangement can, however, not be answered with the experimental set-up employed in this study. However, a recent study showed that Syntaxin-1A and SNAP25 clusters interact at their peripheries through the N-terminal motif of SNAP25[38,56]. We, therefore, speculate that this SNARE motif is also involved in formation of this binary complex. Antiparallel configurations of SNARE assemblies, on the other hand, have been described previously as low-energy states that do not interconvert at an hour time scale[9,10]. An antiparallel config-uration of the SNAP25:Synaptobrevin-2 complex might also explain the decelerated incorporation of Syntaxin-1A; similar to Complexin-1 binding (see above), antiparallel association of a SNARE motif might lock the complex and, therefore, hamper structural re-arrangements.

Considering the different time-scales on which incorporation of the third SNARE into binary pre-assembled complexes occurred, we consider SNAP25 and Syntaxin-1A to be a platform for SNARE assembly readily associating with Synaptobrevin-2. From a functional point of view, prompt formation of the ternary complex is required for spontaneous membrane fusion. However, when this binding platform is not available, formation of a stable complex might take considerably longer presumably due to rearrangements of the protein subunits. Antiparallel configura-tions of the SNARE motifs as discussed above might contribute to extended time scales of complex rearrangements. From these mixing experiments, we conclude that the amount of the available interaction partners does not define the complexes formed but rather the specific interaction partners are responsible for for-mation of stable complexes. Interestingly, stable non-cognate SNARE complexes were obtained when using other members of the Synaptobrevin, SNAP25 and Syntaxin families[58,59]. None-theless, preferred complexes observed in previous and our own

study contain one member of each family suggesting that formation of the functionally active, ternary complex is driven by specificity encoded in the different SNARE motifs.

In terms of complex stability, we found that Synaptobrevin-2 readily dissociates during tandem-mass spectrometry from several (sub-) complexes suggesting least stable incorporation into the helical assemblies. Less stable incorporation of Synaptobrevin-2 into the SNARE complex correlates with its function during SNARE assembly enabling fast association during membrane fusion and facilitating disassembly when the fusion process is completed. This includes partially assembled SNARE complexes in which Synaptobrevin-2 is N-terminally bound to the acceptor complex[60]. The acceptor complex, on the other hand, was previously characterized as a highly stable complex[40] and was also found to be stable in our experiments.

SNARE complex assembly and formation of a fusion pore are essential for successful exocytosis of neurotransmitters from the vesicular lumen into the synaptic cleft. Although several models were discussed, the architecture of the fusion pore remains enigmatic. In vitro fusion experiments showed that, in principle, one SNARE complex is sufficient for membrane fusion[61]; however, other studies found that higher numbers of SNARE complexes are required[62–65]. Multimerisation of the ternary SNARE complex as well as binary complexes was described previously[32]. Likewise, we found that both binary sub-complexes as well as the fully assembled SNARE complex multimerise. Importantly, our models and structural predictions suggest multimerisation of the SNARE complex in an antiparallel manner raising the question, whether multimerisation plays a role during fusion pore formation. Nonetheless, SNARE multimers identified by native MS showed high intensities suggesting that multimerisation is prevalent in solution when using soluble protein constructs. Recent studies suggested an increase in SNARE complex oligomerisation upon Complexin binding[26,46]. These studies revealed a zigzag array of SNARE complexes[46] in which the structural elements of Complexin bind and bridge the SNARE complexes[26]. In our experiments, we clearly observed decreased SNARE multimerisation upon Complexin-1 binding. We assume that Complexin-1 indeed likely plays a regulatory role for SNARE-mediated membrane fusion; however, whether this role is activating or inhibitory remains to be elucidated in future studies.

A recent model reveals binding of the Complexin accessory helix to the membrane-proximal regions of Synaptobrevin-2 and SNAP25 thereby fixing the SNARE complex in an inactive state that is spontaneously released upon arrival of a trigger[66]. Our structural model obtained from cross-linking MS confirms this arrangement of the accessory helix. Importantly, our model includes the N-terminal unstructured region and provides interaction sites with Synaptobrevin-2, SNAP25 and Syntaxin-1A extending our knowledge on the interactions of full-length Complexin-1 with the SNARE complex. A recent study revealed that disassembly of SNARE complexes by the NSF/α-SNAP machinery is prevented by antiparallel binding of Complexin to the SNARE complex[67]. Considering these and our findings, we speculate that binding of Complexin-1 to the assembled ternary SNARE complex represents a control mechanism maintaining the SNARE complex in a pre-active state that can be released when required. This locked pre-active state is protected from disassembly.

From a technical point of view, native MS is the gold standard for determining the protein stoichiometry of protein complexes. This is particularly important when studying protein complexes that differ in the copy number of subunits rather than their composition. Of particular importance for this study, native MS can deal with heterogeneous complex mixtures allowing the identification of complex populations of different intensities which is not possible with most biophysical techniques. These techniques usually report on mixtures of populations while our approach allows to distinguish between defined populations. Importantly, native MS does not require covalent modification or linkage of the subunits, therefore, resembling the natural complexes present in solution. Combining results obtained from native MS and cross-linking MS therefore delivers structural models that reflect the stoichiometry of the proteins and include those regions which are flexible and dynamic and, therefore, difficult to tackle with the classical structural techniques. This is also true for computational predictions, which are often based on high-resolution templates. If high-resolution templates are missing, these predictions often fail integrating flexible and disordered protein regions. Cross-linking MS, on the other hand, provides insights into interactions of unstructured regions facilitating model building. Mutational analysis introducing point mutations at key interaction sites as well as deletion of structural elements will provide additional insights in future experiments.

In their natural environment, the three SNAREs are anchored to the vesicular (Synaptobrevin-2) and the pre-synaptic (Syntaxin-1A and SNAP25) membranes. Employing soluble SNARE constructs in this study allows the unrestricted formation of sub-complexes in various orientations including antiparallel off-pathway complexes. Future studies, structurally characterising potential off-pathway complexes, will provide detailed insights into the mechanism and specificity of SNARE complex formation. Employing full-length proteins reconstituted in membrane mimetics will further restrict the rotational freedom of the proteins in solution and mimic the natural protein environment. However, due to lipid clustering and the comparably prevalent ionisation of lipids, the use of liposomes or nanodiscs complicates the analysis of protein complexes by native MS[68–71] particularly of heterogeneous mixtures of protein complexes as observed here. Continuous methodological advances are therefore required. Note that the effects of lipid membranes onto individual SNARE proteins were studied previously showing that the SNARE motifs remain mostly unstructured and only positively charged residues of the juxtamembrane regions form helical segments[72–74]. These helical segments likely stabilise the proteins in the membrane and position the unstructured SNARE motifs for SNARE zippering[75]. Nonetheless, studying the interactions of SNARE proteins in vivo will provide valuable insights into their interactions as well as unknown intermediates in future studies. The focus of this study, however, is the stoichiometry of potential intermediates and off-pathway complexes. Our results, therefore, contribute to the understanding of SNARE complex assembly and provide the basis for future applications in native-like environments or even in vivo.

## Materials and methods

**Protein variants**. A cysteine-free variant of SNAP25 (SNAP25(CtoS)) in which all cysteine residues were replaced by serine residues, variants of Syntaxin-1A (Stx(1-262)) and Synaptobrevin-2 (Syb(1-96)) lacking their transmembrane helices and full-length Complexin-1 were used. All constructs encode for an N-terminal Hexahistidin-tag (6×His-tag).

**Protein purification**. SNARE variants were expressed in *Escherichia coli* (*E. coli*) BL21 (DE3) cells and expression was induced with 0.4 mM Isopropyl-β-d-1-thio-galactopyranoside overnight at 22 °C in the presence of 30 μg/mL kanamycin. Cells were mechanically lysed in 20 mM HEPES, pH 7.4, 500 mM NaCl, 20 mM Imidazol, 0.1 mM TCEP using a cell disruptor in the presence of protease inhibitor cocktail (Roche) and phenylmethylsulfonyl fluoride. Proteins were isolated from the lysate by immobilized metal affinity chromatography (IMAC). For this, the cell lysate was loaded onto HisTrap HP columns (GE Healthcare) with 20 mM HEPES, pH 7.4, 500 mM NaCl, 20 mM imidazole, 0.1 mM TCEP. 6×His-tagged proteins were eluted from the column using the following concentrations of imidazole: 250 mM for SNAP25 and 375 mM for Syntaxin-1A or Synaptobrevin-2. For 6xHis-tag cleavage, the proteins were dialysed overnight at 4 °C against 20 mM HEPES, pH 7.4, 500 mM NaCl, 20 mM Imidazol, 0.1 mM TCEP containing 50 U thrombin.

6xHis-tags were separated from the proteins by reversed IMAC following the same procedure as described above. The proteins were collected in the flow-through. SNAP25 and Syntaxin-1A were further purified by anion exchange chromatography using HiTrap Q HP columns (GE Healthcare) in 20 mM HEPES, pH 7.4, 50 mM NaCl, 0.1 mM TCEP. SNAP25 was collected in the flow-through and Syntaxin-1A eluted from the column at 550 mM NaCl. Synaptobrevin-2 was further purified by cation exchange chromatography using a HiTrap Q SP column (GE Healthcare) in 20 mM sodium acetate, pH 5.5, 50 mM NaCl, 0.1 mM TCEP and eluted from the column at 200 mM NaCl. Finally, the SNARE proteins were purified by SEC using a HiLoad 16/600 Superdex 75 prep grade column (GE Healthcare) in 20 mM HEPES, pH 7.4, 150 mM KCl, 0.1 mM TCEP, 1 mM EDTA.

Complexin-1 was purified according to the purification protocol described above with few modifications. 20 mM HEPES, pH 7.4, 300 mM NaCl, 0.1 mM TCEP was used for IMAC. Complexin-1 eluted at 300 mM imidazole. Following the reversed IMAC, an additional dialysis was performed to decrease the salt concentration for ion exchange chromatography using 20 mM HEPES, pH 7.4, 50 mM NaCl, 0.1 mM TCEP. Anion exchange chromatography was then performed using 20 mM HEPES, pH 7.4, 0.1 mM TCEP; Complexin-1 eluted at 250 mM NaCl. Complexin-1 was concentrated by filtration through a molecular weight cut-off filter of 10 kDa (Amicon Ultra 15 Ultracell 10k, Merck) exchanging the purification buffer against 20 mM HEPES 7.4, 300 mM NaCl, 0.1 mM TCEP, 1 mM EDTA.

**Complex formation**. Ten to sixty microlitre of the individual SNARE proteins in purification buffer were mixed in varying ratios (see figure legends for details). For SNARE complex formation, SNAP25 and Synaptobrevin-2 were mixed in a 1:1 ratio followed by equimolar addition of Syntaxin-1A to prevent formation of the dead end complex[30]. Mixed samples were incubated overnight at 4 °C. For rearranging of preassembled complexes, additional interaction partners were added and directly analysed.

**Gel electrophoresis and western blotting**. (Cross-linked) Proteins were separated by gel electrophoresis on Bis-Tris gels (4 - 12%) using the NuPAGE system (Thermo Scientific). Electrophoresis was performed at constant voltage of 200 V for 35 min. The SeeBlue Plus 2 Pre-Stained Protein Marker (Thermo Scientific) was used as molecular weight protein marker. Protein gels were stained with Coomassie using InstantBlue Protein Stain (Expedeon).

For western blotting, (cross-linked) proteins were electrophoretically transferred onto a nitrocellulose membrane (Amersham Protran Nitrocellulose Blotting membrane, GE Healthcare) for 2 h at 50 mA. The blotting membrane was blocked with 5% (w/v) milk powder in phosphate-buffered saline, 0.02% (v/v) Tween 20 followed by incubation in phosphate-buffered saline, 0.02% (v/v) Tween 20, 1% (w/v) BSA containing anti-SNAP25 clone 71.1 (1:10.000), anti-Syntaxin-1A clone 78.3 (1:10.000) or anti-Complexin-1/2 (1:1000) antibodies (SynapticSystems) overnight at 4 °C. Following an additional washing step with phosphate-buffered saline, 0.02% (v/v) Tween 20, the membrane was incubated in phosphate-buffered saline, 0.02% (v/v) Tween 20, 1% (w/v) BSA containing anti-mouse (1:10.000; for anti-SNAP25 clone 71.1 and anti-Syntaxin-1A clone 78.3) or anti-rabbit (1:10.000; for anti-Complexin-1/2) secondary antibodies for at least 1 h at room temperature. The western blot was developed using Pierce ECL Western Blotting Substrate (Thermo Scientific) and the resulting chemiluminescence was detected using a Luminescent Image Analyzer (LAS 4000, Fujifilm).

**Chemical cross-linking**. For optimizing the cross-linker concentration, 10 μM SNAP25, Syntaxin-1A and Complexin-1 were cross-linked with 0.1-1.5 mM BS3 for 1 h at 25 °C and 350 rpm. For identification of cross-linking sites, 10 μM of the individual SNAREs or 50 μM of Complexin-1 were cross-linked with 0.5 mM or 1.5 mM BS3 in 100 μL.

**Proteolysis**. For hydrolysis of proteins in solution, (cross-linked) proteins were precipitated with ethanol. For this, protein samples were diluted with water to a final volume of 200 μL and precipitated by addition of 20 μL 3 M sodium acetate, pH 5.3 as well as 600 μL ice-cold 100 % (v/v) ethanol. Following incubation overnight, the proteins were pelleted, washed with 80 % (v/v) ethanol and dried using a vacuum centrifuge. Resulting protein pellets of individual proteins were suspended in 10 μL 1% (v/v) RapiGest (Waters) in 25 mM NH₄HCO₃, pH 8.5. The protein pellet of the ternary SNARE complex was dissolved in 10 μL 8 M urea. Disulphide bonds were reduced with 10 μL 50 mM dithiothreitol in 25 mM NH₄HCO₃, pH 8.5 for 1 h at 37 °C. Subsequently, free cysteine residues were alkylated with 10 μL 100 mM iodoacetamide in 25 mM NH₄HCO₃, pH 8.5 for 1 h at 37 °C in the dark. Proteins were hydrolysed in a total volume of 100 μL with trypsin (Promega) in 25 mM NH₄HCO₃, pH 8.5, at an enzyme:protein ratio of 1:20, followed by incubation overnight at 37 °C. Following proteolysis, RapiGest was hydrolysed by addition of 20 μL 5% (v/v) trifluoroacetic acid (TFA) followed by 2 h incubation. Tryptic peptides were then collected after centrifugation and dried in a vacuum centrifuge.

**Enrichment of cross-linked peptide pairs by SEC**. For this, dried peptides were dissolved in 60 μL 30% (v/v) acetonitrile (ACN), 0.1% (v/v) TFA and isocratically separated on a Superdex peptide column 3.2/300 GL (GE healthcare) using an Äkta *pure* system at a flow rate of 50 μL/min. Fractions containing cross-linked peptide pairs were collected, peptides were dried using a vacuum centrifuge and subsequently analysed by LC-MS/MS.

**LC-MS/MS analysis**. LC-MS/MS analysis was performed by reversed-phase liquid chromatography using a DionexUltiMate 3000 RSLC nano System (Thermo Scientific; mobile phase A: 0.1 % (v/v) formic acid (FA); mobile phase B: 80 % (v/v) ACN, 0.1 % (v/v) FA) coupled to a Q Exactive Plus Hybrid Quadrupol-Orbitrap mass spectrometer (Thermo Scientific). For protein identification and identification of cross-linking sites, peptides were dissolved in 2% (v/v) ACN, 0.1% (v/v) FA. The peptides were loaded onto a trap column (Acclaim PepMap 100 C18-LC column, 300 μm I.D., particle size 5 μm; Thermo Scientific) and separated on an analytical column (Acclaim PepMap 100 C18-LC column, 75 μm I.D., particle size 3 μm; Thermo Scientific) at a flow rate of 300 nL/min. Samples were separated using a gradient of 4-90% mobile phase B in 69 min for protein identification and 99 min for cross-linking analysis. The gradient was adjusted depending on the elution of cross-linked peptide pairs in early, middle or late fractions of the SEC.

The following parameters were used for MS data acquisition: spray voltage, 2.8 kV; capillary temperature, 275 °C; data-dependent mode. Survey full scans were acquired in the Orbitrap (m/z 350-1600) with a resolution of 70.000 and an automatic gain control (AGC) target of 3e6. The 20 most intense ions with charge states of 2+ to 8 + (for protein identification) and 3+ to 8 + (for cross-linking analysis) were selected and fragmented in the HCD cell at an AGC target of 1e5 and a normalized collision energy of 30%. Previously selected ions were dynamically excluded for 30 s. The lock mass option (lock mass m/z 445.120025[76]) was enabled.

**Database searching and data analysis**. For protein identification, raw data were searched against a database including the *E. coli* proteome as well as the corresponding target proteins using MaxQuant (v.1.6.17) software[77]. The following search parameters were applied: enzyme, trypsin; missed cleavage-sites, 2; variable modifications, carbamidomethylation (cysteine), oxidation (methionine) and acetylation (N-terminus); mass accuracy, 20 ppm for precursor ions and 4.5 ppm fragment ions; false discovery rate, 0.01.

For identification of cross-linked peptide pairs, raw-data were searched against a minimized database using pLink (v.2.3.9)[78]. The following search parameters were applied: enzyme, trypsin; missed cleavage sites, 3; peptide mass, 600-6000; peptide length, 6-60; precursor and fragment tolerance + /- 20 ppm; fixed modification, carbamidomethylation (cysteine); variable modification, oxidation (methionine); fragmentation, HCD; false discovery rate, 0.05; cross-linker, BS3 (cross-links N-termini and lysine, serine, threonine and tyrosine residues). Mass spectra of potential cross-links were manually validated. Results tables were processed using CroCo software[79] and identified cross-links were visualized using xVis software[80]. For validation, cross-links were mapped into the high-resolution structure of the SNARE complex (PDB ID: 1SFC)[5] and an AlphaFold2 prediction of the SNARE:Complexin-1 complex using UCSF Chimera[81] and Xlink Analyzer[82]. For AlphaFold prediction, ColabFold[47] was used. The following parameters were employed: query_sequence, protein sequences of SNAP25(CtoS), Syb(1-96), Stx(1-262) and full-length Complexin-1; template_mode, none; msa_mode, MMseqs2 (UniRef and Environmental); pair_mode, unpaired and paired; model_type, AlphaFold2-ptm and AlphaFold-multimer-v2; num_recycles, 3. To validate the antiparallel orientation of SNARE complexes observed by visualising cross-links in the crystal structure, multimers of the SNARE complex were predicted using ColabFold (see above) by implementing the protein sequences twice during prediction.

**Model building**. For visualisation of cross-links identified in the SNARE:Complexin-1 complex, a model based on the complete SNARE complex and full-length Complexin-1 was created using PyMOL[83]. For this, a high-resolution structure of the SNARE complex (PDB ID: 1SFC)[5] was aligned with an incomplete structure of the SNARE complex lacking structures for Synaptobrevin-2 but containing the central helix of Complexin-1 (PDB ID: 1KIL)[23]. Next, a structure of full-length Complexin-1 was predicted using AlphaFold[48]. The central helix of this predicted structure was then aligned with the central helix of Complexin-1 in the SNARE complex template (see above). Finally, an available high-resolution structure of the Habc domain of Syntaxin-1 (PDB ID: 1EZ3) was manually added to the arrangement[50]. For cross-link visualisation, missing structural elements such as the flexible linker of SNAP25, the C-terminus of Syntaxin-1A and the flexible linker of Syntaxin-1A connecting the SNARE motif and the Habc domain were added manually as grey lines.

**Native MS**. Native MS experiments were performed on a Waters Micromass Q-ToF Ultima mass spectrometer modified for transmission of high masses[84]. For this, the storage buffer of 20 μL protein solution was exchanged to 200 mM ammonium acetate using Micro Bio-Spin 6 gel filtration columns (BioRad) or Vivaspin 500 filtration units with a molecular weight cut-off of 10 kDa (Sartorius AG). The proteins were then diluted to concentrations of 5-30 μM. 4 μL of the protein (complexes) were loaded into gold-coated glass capillaries prepared in-

house[76] and directly introduced into the mass spectrometer. The following parameters were used for data acquisition: capillary voltage, 1.3 – 1.7 kV; sample cone voltage, 80 V; RF lense voltage, 80 V; collision voltage, 10 – 50 V. Mass spectra were processed using MassLynx 4.1 (Waters), externally calibrated using caesium iodide solution (100 mg/mL) and analysed using MassLynx 4.1, Massign software (version 11/14/2014)[85] and an in-house written deconvolution macro. All theoretically calculated and experimentally determined masses are given in Supplementary Table 2.

**CD spectroscopy**. CD spectroscopy was performed using a Jasco J-815 CD spectrometer. For this, 30-60 μM of the individual proteins and 60-90 μM of the binary and ternary complexes were used. CD spectra were acquired over a range of wavelength of 195-260 nm with 0.1 nm step size at 10 °C using a 0.01 cm or 0.001 cm path length quartz cuvette. For each measurement, the baseline of the buffer was subtracted and 32 scans were averaged. All measurements were performed in 20 mM HEPES, pH 7.4, 150 mM KCl, 0.1 mM TCEP and 1 mM EDTA. To evaluate structural changes resulting from complex formation, theoretical spectra were calculated from the CD spectra of the individual SNARE proteins. Therefore, the molar ellipticity in (deg cm$^2$ dmol$^{-1}$) was calculated ($[\Theta] = \Theta / (10 \cdot c \cdot d)$) where $\Theta$ is the observed ellipticity in mdeg, c is the concentration in mol·l$^{-1}$ and d is the path length in cm) for the individual proteins and summed ($[\Theta]_{th} = [\Theta]_1 + [\Theta]_2$) taking the employed protein ratios into account. The data were smoothed using the Savitzky-Golay algorithm with a smooth window of 40.

**Statistics and Reproducibility**. Purification of the proteins was confirmed by mass spectrometry once. A false discovery rate of 1% was applied during database search (MaxQuant). All native MS experiments were performed at least three times. Protein complexes were individually mixed in separate experiments. Representative mass spectra are shown. The standard deviation for the experimentally determined masses is given (Supplementary Table 2). All cross-linking experiments were performed three times using individual proteins or protein complexes mixed in individual experiments (gel-based visualisation and MS-based identification). A false discovery rate of 5% was applied during database search (pLink). CD measurements were performed at least in duplicates.

**Reporting summary**. Further information on research design is available in the Nature Portfolio Reporting Summary linked to this article.

# Data availability

All MS raw files and the corresponding results files including databases were deposited to the ProteomeXchange Consortium (www.proteomexchange.org) via the PRIDE[86] partner repository with the dataset identifier PXD030619. Uncropped gel and blot images are available in Supplementary Figure 1. All other data are available from the corresponding author upon reasonable request.

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

## Acknowledgements

We thank Reinhard Jahn (MPI for Multidisciplinary Sciences) for critically reading our manuscript and for providing plasmids of SNAP25, Syntaxin-1A, Synaptobrevin-2 and Complexin-1. We also thank Sonja Pribicevic (MPI for Multidisciplinary Sciences) and Ángel Pérez-Lara (University of Granada) for helpful discussion and Ralph Golbik (Martin Luther University Halle-Wittenberg) for help with interpretation of CD spectra. We acknowledge funding from the Federal Ministry for Education and Research (BMBF, 03Z22HN22), the European Regional Development Funds (EFRE, ZS/2016/04/78115), the MLU Halle-Wittenberg, the JGU Mainz and the German Research Foundation (DFG, project number 391498659, RTG 2467 "Intrinsically Disordered Proteins – Molecular Principles, Cellular Functions, and Diseases").

## Author contributions

J.H. performed all experiments and analysed the data. J.H. and C.S. wrote the manuscript. C.S. supervised the work and guided the research.

## Funding

## Competing interests

The authors declare no competing interests.
