## [Peer Review File · Communications Biology]

Reviewers' comments:

Reviewer #1 (Remarks to the Author):

Reviewer's comments on the manuscript by Hesselbarth and Schmidt "A Roadmap for SNARE Complex Assembly: Mass Spectrometry Uncovers Protein Interactions and Stoichiometry of Intermediates and Sub-complexes"

The formation of the SNARE complex is of the utmost importance for fusion of neurotransmitter-filled synaptic vesicles with the plasma membrane upon arrival of an action potential. The mechanism of SNARE complex formation has been extensively studied *in vitro* and *in vivo* including the underlying kinetics, the structure of the SNARE complex, and the interaction with other proteins. Formation of the complex during membrane fusion *in vivo* is also dependent on the function of additional proteins, among others Complex 1.

In their manuscript, Hesselbarth and Schmidt have investigated the assembly of SNARE complex and its interaction with the protein complexin-1 by mass spectrometry (MS), a method that had not yet been applied to this particular topic. The authors conducted MS on reconstituted assembled SNARE complexes and their subcomplexes. In addition, they monitored the intra- and inter-protein interactions in and among the single SNARE proteins as well as the SNARE-Complexin-1 complex; to do this they used crosslinking combined with MS. On the basis of their experiments the authors developed an assembly map that includes the interactions of the SNARE complex, with and without Complexin-1 protein, and identifies off-pathway complexes and their reassembly; this goes beyond maps derived in earlier published studies.

Overall, I am in great favour of the study presented here for the following reasons: (1) The MS-based investigations reveal the exact protein stoichiometries of SNARE subcomplexes and of the fully assembled SNARE complex with and without Complexin-1. The study is another prime example demonstrating that mass spectrometry of intact protein complexes is a highly precise method to tackle the (in practice very difficult) task of determining protein stoichiometries in complexes when no 3D structures are available. In fact, the observed heterogeneity of complexes can only be tackled with this technique. Hesselbarth and Schmidt's mass spectra of the intact proteins and complexes are of exceptionally high quality. (2) The crosslinking results reveal the conformations of the proteins, both alone and when assembled in a complex. The molecular interactions of Complexin-1 with the SNARE, as presented in this work, extend the range of available 3D structures and provide yet another example of how 3D models can be generated by combining protein crosslinking and AlphaFold predictions. (3) I consider that the *in vitro* assembly system established here, including the manner in which complex formation is monitored (i.e., by native MS) has a great potential for investigating other proteins that act on SNAREs in assembly and/or disassembly and will hence provide the basis for future *in vivo* studies. In addition, the identification of stable subcomplexes and off-pathway complexes expands our knowledge on SNARE complex formation in general; for instance, the formation of non-cognate SNARE complexes is still enigmatic and future studies might benefit from the approach introduced here.

I consider that this manuscript is very well-suited for publication in *Communications Biology*. I would nonetheless like to list some points which the authors may address and which may further increase the value of their study:

1. The classical view of a SNARE complex and its subcomplexes is a four-helix bundle. Here Hesselbarth and Schmidt present complexes with a 1:2 stoichiometry (SNAP25/Synaptobrevin) which does not match with the four-helix bundle. However, the authors show by CD spectroscopy that helical structures are formed. Are the authors able to tell whether a helical bundle was formed? Assuming a helical bundle similar to the SNARE complex one Q-SNARE helix should not be included; are the authors able to tell which Q SNARE helix is not within the four helical bundle?
2. The possible physiological significance a SNAP25-Synaptobrevin binary complex is not entirely clear to me. The authors would do well to explain this in more detail, in the context of membrane fusion and SV recycling.
3. The fact that the authors observed that the "dead-end" (kinetically trapped) complex is actually not a dead-end (kinetically trapped) complex at all, and can assemble very rapidly with Synaptobrevin

into a trimeric SNARE, is very interesting. The authors discuss this in the context of their observation that the dead-end complex was observed in liposome fusion experiments. Do the authors think that this complex might also exist *in vivo*? Would such a 4Q complex not also form when several SNAP25-Syntaxin-1 proteins are clustered together on the plasma membrane? Maybe the authors can discuss this in more detail, also with respect to the other proteins that are shown to mediate vesicle fusion in addition to the SNAREs. Moreover, charge +20 is the preferred one of the dead-end complex in Figure 2A, but in Figure S2 (and S5) it has the charge state +19. Might this reflect certain conformational differences of the dead-end complexes?

4. The MS experiments with Complexin 1 show that it prevents multimerisation of a trimeric SNARE complex. Complexin 1 interacts with Syntaxin 1/Synaptobrevin and also with the 3Q:1R trimeric SNARE. From the crosslinking experiments of 3Q:1R and 3Q:1R:Complex 1 SNARE however (Figure S9), it seems that Complexin-1 prevents direct interaction between SNAP25 and Syntaxin 1 and might also replace interactions with Synaptobrevin. It is not entirely clear whether the authors have removed the crosslinks of the canonical SNARE from panel B in Figure S9 for the sake of simplicity. In my opinion, a crosslinking map that shows all the crosslinks in the SNARE-Complexin-1 complex in a distinguishable manner would be beneficial.

5. Line 244: Which ratio of Synaptobrevin was added?

In summary, I strongly recommend the publication of this manuscript *Communications Biology*. However, it would improve the value of the author's study if they can address as many as possible of the points listed above.

Reviewer #2 (Remarks to the Author):

The present manuscript gives a welcome overview of SNARE complex assembly, by using mainly native MS to characterize different subcomplexes of pairwise and higher order interactions between constituents. Hence the main reported outputs are also stoichiometries, with some crosslinking data included as well. Structural conclusions are more difficult to draw here apart from the SNARE:Complexin-1 (Figure 6) mostly due to the absence of lipids and the use of truncated (soluble) versions of the proteins, but nevertheless some interesting insights can be gained.

I recommend this paper for publication subject to addressing some points which imho will increase the value of the reported data to the non-expert reader.

- Showing a model of current understanding of SNARE complex assembly in the context of their functional role would be helpful, to accompany the introduction

- Please clarify what you mean by parallel/antiparallel here

- What is the SNAP antibody specific for?

- How do you tell intra- and intermolecular crosslinks apart here?

- I would assume that SEC enriches xlinked peptides here rather than (fully) separating them - can you please clarify?

- Can you comment on the discrepancy seen in the gels and native MS regarding the homo-oligomerization behaviour?

- Fig 2B shows considerable amount of remaining monomer, is the affinity for binding rather weak or what other explanation could you suggest?

- Gas-phase dissociation products (p10) are only explained later, for the non-expert reader this is not

obvious

- You might want to consider adding some simple bar graphs or similar to the native MS spectra to visualize the total relative intensities of different complexes across charge states

- I would suggest a broader discussion of how observations such as oligomerization and apparent incomplete binding may be due to the use of soluble forms in the absence of lipids - what is known about the effect lipids might have on these interactions?

- Regarding "kinetically trapped species", have you compared short with long incubation times for some of these complexes to see if reorganisation is needed and slow? Does the order in which components are added to each other matter?

- Volumes should be reported using L (not l)

Reviewer #3 (Remarks to the Author):

The paper is a comprehensive analysis of neuronal SNARE protein complexes in the presence and absence of the regulatory protein Complexin by cross-linking and native mass spectrometry. Using these methods, the authors have identified and characterised several intermediate SNARE complexes as well as studied the role of Complexin on stabilising the SNARE complex. The work reveals the complexities of SNARE complex assembly that can be followed up by further analysis by other methods or in a native environment.

SNARE mediated membrane fusion is a complex and highly regulated process. The manuscript highlights the ability of using proteomics to complement structural and other biophysical methods to tease apart the multiple interactions that drive vesicle fusion. They have also shown its utility in mapping out interactions within flexible regions that are not easily resolved by structural techniques.

1. The biological relevance of the intermediate complexes are somewhat ambiguous given that soluble proteins are used rather than those in a native membrane environment. The manuscript will benefit from a short discussion of this point.

2. With the modelling some parts of the proteins were added manually. Have the authors considered using alphafold multimer to model the SNARE-Complexin complex and assessing how well their crosslinks agree with the model?

3. How do the authors reconcile their observations with the fast kinetics associated with vesicle fusion?

4. Further validation of their observations through mutational analysis will be beneficial and perhaps warrants a comment in the discussion.

Reviewers' comments:

We thank the reviewers for their thorough review and the valuable suggestions; we modified our manuscript accordingly. We hope that the reviewers appreciate the modifications that we made and that they find our manuscript now acceptable for publication.

Reviewer #1 (Remarks to the Author):

Reviewer's comments on the manuscript by Hesselbarth and Schmidt "A Roadmap for SNARE Complex Assembly: Mass Spectrometry Uncovers Protein Interactions and Stoichiometry of Intermediates and Sub-complexes"

The formation of the SNARE complex is of the utmost importance for fusion of neurotransmitter-filled synaptic vesicles with the plasma membrane upon arrival of an action potential. The mechanism of SNARE complex formation has been extensively studied in vitro and in vivo including the underlying kinetics, the structure of the SNARE complex, and the interaction with other proteins. Formation of the complex during membrane fusion in vivo is also dependent on the function of additional proteins, among others Complex 1.

In their manuscript, Hesselbarth and Schmidt have investigated the assembly of SNARE complex and its interaction with the protein complexin-1 by mass spectrometry (MS), a method that had not yet been applied to this particular topic. The authors conducted MS on reconstituted assembled SNARE complexes and their subcomplexes. In addition, they monitored the intra- and inter-protein interactions in and among the single SNARE proteins as well as the SNARE-Complexin-1 complex; to do this they used crosslinking combined with MS. On the basis of their experiments the authors developed an assembly map that includes the interactions of the SNARE complex, with and without Complexin-1 protein, and identifies off-pathway complexes and their reassembly; this goes beyond maps derived in earlier published studies.

Overall, I am in great favour of the study presented here for the following reasons: (1) The MS-based investigations reveal the exact protein stoichiometries of SNARE subcomplexes and of the fully assembled SNARE complex with and without Complexin-1. The study is another prime example demonstrating that mass spectrometry of intact protein complexes is a highly precise method to tackle the (in practice very difficult) task of determining protein stoichiometries in complexes when no 3D structures are available. In fact, the observed heterogeneity of complexes can only be tackled with this technique. Hesselbarth and Schmidt's mass spectra of the intact proteins and complexes are of exceptionally high quality. (2) The crosslinking results reveal the conformations of the

proteins, both alone and when assembled in a complex. The molecular interactions of Complexin-1 with the SNARE, as presented in this work, extend the range of available 3D structures and provide yet another example of how 3D models can be generated by combining protein crosslinking and AlphaFold predictions. (3) I consider that the in vitro assembly system established here, including the manner in which complex formation is monitored (i.e., by native MS) has a great potential for investigating other proteins that act on SNAREs in assembly and/or disassembly and will hence provide the basis for future in vivo studies. In addition, the identification of stable subcomplexes and off-pathway complexes expands our knowledge on SNARE complex formation in general; for instance, the formation of non-cognate SNARE complexes is still enigmatic and future studies might benefit from the approach introduced here.

I consider that this manuscript is very well-suited for publication in Communications Biology. I would nonetheless like to list some points which the authors may address and which may further increase the value of their study:

1. The classical view of a SNARE complex and its subcomplexes is a four-helix bundle. Here Hesselbarth and Schmidt present complexes with a 1:2 stoichiometry (SNAP25/Synaptobrevin) which does not match with the four-helix bundle. However, the authors show by CD spectroscopy that helical structures are formed. Are the authors able to tell whether a helical bundle was formed? Assuming a helical bundle similar to the SNARE complex one Q-SNARE helix should not be included; are the authors able to tell which Q SNARE helix is not within the four helical bundle?

Unfortunately, we cannot tell whether (or not) a four-helix bundle was formed. Our data only inform on the content of secondary structure and we, indeed, identify an increase in helical content. However, previous studies reported on interactions between individual SNARE proteins that are similar to those formed within the four-helix bundle.

The question which SNARE motif of SNAP25 is engaged in complex formation is indeed very interesting. However, with the data obtained here, we are not able to answer this question. Nonetheless, a previous study showed that Syntaxin-1A and SNAP25 clusters interact through the N-terminal SNARE motif of SNAP25. We therefore speculate whether this SNARE motif is also involved in complex formation in the Synaptobrevin-2:SNAP25 complex. We have added a comment on this to our revised manuscript (I. 653-663):

“Assuming a four-helical bundle similar to the fully assembled ternary SNARE complex, the additional fifth SNARE motif is likely unfolded and not involved in formation of the complex. This unstructured loop might interact with the binding

interfaces of the complex or sterically hinder rearrangements. The question whether the N- or the C-terminal SNARE motif of SNAP25 is engaged in the helical arrangement can, however, not be answered with the experimental set-up employed in this study. However, a recent study showed that Syntaxin-1A and SNAP25 clusters interact at their peripheries through the N-terminal motif of SNAP25^{38,56}. We, therefore, speculate that this SNARE motif is also involved in formation of this binary complex. Antiparallel configurations of SNARE assemblies, on the other hand, have been described previously as low-energy states that do not interconvert at an hour time scale.”

2. The possible physiological significance a SNAP25-Synaptobrevin binary complex is not entirely clear to me. The authors would do well to explain this in more detail, in the context of membrane fusion and SV recycling.

The SNAP25:Synaptobrevin-2 complex was previously described for PC12 cells as the first step of SNARE-mediated membrane fusion. However, the physiological relevance of this complex was controversially discussed for various species in several other studies. We have added this information to our revised manuscript to better provide the rationale behind studying this complex. The corresponding section now reads (l. 273-278):

“We next studied complex formation between SNAP25 and Synaptobrevin-2. Formation of a complex between SNAP25 and Synaptobrevin-2 was controversially discussed in several previous studies. While the SNAP25:Synaptobrevin-2 complex was suggested to represent the first step of membrane fusion in PC12 cells³⁸, other studies did not observe stable complexes³⁹ or only weak interactions^{40,41}. We, therefore, mixed the two SNAREs in a 1:1 molar ratio and explored formation of complexes by native MS.”

3. The fact that the authors observed that the “dead-end” (kinetically trapped) complex is actually not a dead-end (kinetically trapped) complex at all, and can assemble very rapidly with Synaptobrevin into a trimeric SNARE, is very interesting. The authors discuss this in the context of their observation that the dead-end complex was observed in liposome fusion experiments. Do the authors think that this complex might also exist in vivo? Would such a 4Q complex not also form when several SNAP25-Syntaxin-1 proteins are clustered together on the plasma membrane? Maybe the authors can discuss this in more detail, also with respect to the other proteins that are shown to mediate vesicle fusion in addition to the SNAREs.

We agree with the reviewers that clustering of Syntaxin-1A and SNAP25 proteins was described previously, however, interactions between the two proteins were only observed at the peripheries of the clusters. Note that additional regulators also play an important role in formation (and disassembly) of the acceptor complex. We included more details on this into the discussion of our manuscript (I. 625-637):

“...An equilibrium between the 2:1 ‘dead end acceptor complex’ and the 1:1 ‘active acceptor complex’, which were both observed in our mass spectra, might accelerate complex’ rearrangements or disassembly-reassembly processes. Similar assumptions were made previously when studying the mesoscale organisation of the plasma membrane in PC12 cells⁵⁶. This study revealed co-existing tight clusters of alpha-helical Syntaxin-1A bundles and diffuse clusters of SNAP25 interacting only at their peripheries. Diffuse SNAP25 clusters are recruited to Syntaxin-1A clusters through alpha-helical interactions of the N-terminal SNARE motif of SNAP25 showing Syntaxin-1A:SNAP25 complexes only at the interfaces of the clusters. Higher abundance of SNAP25 in the membranes, as observed in PC12 cells, prevents formation of the ‘dead end acceptor complex’ and drives formation of 1:1 ‘active acceptor complexes’⁵⁶. However, the abundance of SNAP25 differs between species and additional regulators such as Munc18 and Munc13 regulate formation of the acceptor complex¹²⁻¹⁴.

Moreover, charge +20 is the preferred one of the dead-end complex in Figure 2A, but in Figure S2 (and S5) it has the charge state +19. Might this reflect certain conformational differences of the dead-end complexes?

A difference of one charge between two mass spectra is usually not reflecting a meaningful difference. Note that charge states can be easily manipulated, for instance, by the position of the needle with respect to the source orifice. We, therefore, do not specifically comment on this finding.

4. The MS experiments with Complexin 1 show that it prevents multimerisation of a trimeric SNARE complex. Complexin 1 interacts with Syntaxin 1/Synaptobrevin and also with the 3Q:1R trimeric SNARE. From the crosslinking experiments of 3Q:1R and 3Q:1R:Complex 1 SNARE however (Figure S9), it seems that Complexin-1 prevents direct interaction between SNAP25 and Syntaxin 1 and might also replace interactions with Synaptobrevin. It is not entirely clear whether the authors have removed the crosslinks of the canonical SNARE from panel B in Figure S9 for the sake of simplicity. In

my opinion, a crosslinking map that shows all the crosslinks in the SNARE-Complexin-1 complex in a distinguishable manner would be beneficial.

We apologise for this confusion. In this study, only the SNARE:Complexin-1 complex was studied by chemical cross-linking. For simplicity, we showed in Figure S10 cross-links observed between the SNARE proteins and cross-links observed between the SNAREs and Complexin-1 in two separate plots. To avoid confusion, we now show the full set of observed cross-links and visualise cross-links observed in the SNARE complex and cross-links observed between the SNAREs and Complexin-1 in different colours (**Supplementary Figure 10**, see below).

5. Line 244: Which ratio of Synaptobrevin was added?

Synaptobrevin-2 was added at a 1.2 molar excess. We added this information to the revised manuscript. It now reads (l. 242-245):

“To interrogate whether the 1:2 SNAP25:Syntaxin-1A complex is kinetically trapped as suggested for the membrane-bound complex, we added a 1.2 molar excess of Synaptobrevin-2, the naturally preferred interaction partner of the ‘acceptor complex’, and followed complex formation by native MS (Figure 2b).”

In summary, I strongly recommend the publication of this manuscript Communications Biology.

However, it would improve the value of the author's study if they can address as many as possible of the points listed above.

Reviewer #2 (Remarks to the Author):

The present manuscript gives a welcome overview of SNARE complex assembly, by using mainly native MS to characterize different subcomplexes of pairwise and higher order interactions between constituents. Hence the main reported outputs are also stoichiometries, with some crosslinking data included as well. Structural conclusions are more difficult to draw here apart from the SNARE:Complexin-1 (Figure 6) mostly due to the absence of lipids and the use of truncated (soluble) versions of the proteins, but nevertheless some interesting insights can be gained.

I recommend this paper for publication subject to addressing some points which imho will increase the value of the reported data to the non-expert reader.

6. Showing a model of current understanding of SNARE complex assembly in the context of their functional role would be helpful, to accompany the introduction

We thank the reviewers for this suggestion; however, we refrain from presenting a model of SNARE assembly for the following reasons: (i) This type of background information would be more suited for a review article. In fact, several recent review articles on SNARE proteins provide an overview on the current knowledge. We cannot cover the depth of these review articles in one assembly pathway. (ii) While there is a current understanding on SNARE complex formation, many details are still under debate. This is particularly true for the regulators of SNARE assembly (such as Complexin-1). As an example, the recent review by Bao and co-workers (Zhang et al (2022) Crit Rev Biochem Mol Biol) presents two alternative assembly pathways. The assembly pathways that are available mostly differ depending on the regulatory factors included. It is therefore difficult to present one commonly accepted assembly pathway. (iii) We provide additional information on possible SNARE intermediates. We summarise our findings in our own assembly pathway (**Figure 7**). We feel that this pathway is better suited for this research article.

7. Please clarify what you mean by parallel/antiparallel here

The SNARE complex involves formation of a four-helix bundle. Parallel and antiparallel configurations, therefore, refer to the parallel or antiparallel alignment of the four helices. Accordingly, in parallel configuration, all helices align from the N- to C-termini, and in antiparallel configurations, the helices align from their N- to C-termini and C- to N-termini. This term is commonly used in structural biology. To avoid confusion with the direction of

zippering, we do not explicitly mention alignment directions, however we have modified the text as follows (l. 71-73):

“In addition to the parallel assembly of the **alpha-helices of the** functionally active ternary SNARE complex, antiparallel configurations have been described that spontaneously assemble...”

8. What is the SNAP antibody specific for?

We used specific antibodies for all proteins used in this study, namely Synaptobrevin-2, Syntaxin-1A, SNAP25 and Complexin-1. The “SNAP antibody” is specific against SNAP25. This is stated in the main text of the manuscript as well as the Methods section. Searching our manuscript, we do not find the term “SNAP antibody”, however, we assume that the reviewers refer to the SNAP25 antibody. Detailed information on the antibody is also provided in the Methods section: anti-SNAP25 clone 71.1 (Synaptic Systems).

9. How do you tell intra- and intermolecular crosslinks apart here?

Intermolecular cross-links refer to cross-links between two proteins. These two proteins are either different proteins or two copies of the same proteins. Intra-molecular cross-links originate from the same proteins. Note that it is generally difficult to distinguish between intra- and intermolecular cross-links of the same protein (i.e. between those cross-links that originate from the same copy and those cross-links that originate from two copies of the same protein). If overlapping sequences of the same protein are observed in a cross-linked peptide pair, the two peptides must originate from two copies of the protein. If the two peptides have different sequences, it is impossible to know whether the two peptides originate from one or two copies of the protein. In this study, cross-linking of the individual SNAREs delivered several inter-molecular cross-links that included overlapping peptide sequences (**Figure 1**). Cross-links observed in the SNARE complex were identified as intra- or intermolecular cross-links by visualising them in the available high-resolution structure (**Supplementary Figure 11**); in this example, some cross-links only satisfied the distance restraint by including a second SNARE complex in the calculation. This agrees well with our finding that the SNARE complex oligomerises.

10. I would assume that SEC enriches xlinked peptides here rather than (fully) separating them - can you please clarify?

This is correct. We apologise for this confusion. Cross-linked peptide pairs are enriched by SEC. We have modified the text accordingly (l. 145-149). It now reads (changes highlighted in red):

“Note that cross-linked peptides were enriched from a mixture of cross-linked and linear peptides by peptide SEC during sample preparation (see Materials and Methods); co-elution during HPLC and gas-phase interactions of linear and loop-linked peptides resulting in the same molecular mass as a cross-linked peptide pair³⁶ can therefore be neglected.”

11. Can you comment on the discrepancy seen in the gels and native MS regarding the homo-oligomerization behaviour?

We actually feel that the cross-linking and native MS experiments agree very well. The only difference between the two methods is the maximum oligomeric state. These differences can be explained by the fact that gel electrophoresis (i.e. Coomassie staining) and Western blotting are more sensitive than native MS and that the conditions used in native MS experiments (buffer exchange, ionisation etc) might also affect the degree of oligomerisation. Accordingly, additional peaks in the mass spectra indicate the presence of higher oligomers, however, these peaks are of low intensity and were therefore not fully assigned. We speculate that unspecific multimerization of the SNAREs is stabilising the proteins in the absence of interactions partners (see Discussion section). We have added the following statement to the discussion section (l. 597-599):

“Differences in the oligomeric state observed by gel electrophoresis/western blotting or native mass spectrometry are likely attributed to the different sensitivity of the methods.”

11. Fig 2B shows considerable amount of remaining monomer, is the affinity for binding rather weak or what other explanation could you suggest?

Synaptobrevin-2 was added at a 1.2 molar excess when compared with the other two proteins (see Figure legend). We therefore provide an excess of Synaptobrevin-2, which is most likely the reason for the presence of the monomeric protein. This information was added to the revised manuscript as also requested by Reviewer 1 (see above, comment #5). In addition, the monomeric protein (Synaptobrevin-2) is disordered in the absence of

interaction partners and ionises better than the fully assembled SNARE complex (this is also true for the other monomeric proteins, e.g. SNAP25 in Figure 3A).

12. Gas-phase dissociation products (p10) are only explained later, for the non-expert reader this is not obvious

We thank the reviewers for this comment. We have modified the text so that the non-expert reader can follow our conclusions. The text now reads (l. 287-291):

“Note that the SNAP25 dimer showed lower charge states than the 2:1 SNAP25:Synaptobrevin-2 complex. In some cases, labile complexes dissociate in the gas-phase resulting in a highly charged dissociated monomer and a “stripped complex” carrying less charges; we suppose that the SNAP25 dimer forms during gas-phase dissociation of the 2:1 complex rather than in solution.”

13. You might want to consider adding some simple bar graphs or similar to the native MS spectra to visualize the total relative intensities of different complexes across charge states

We thank the reviewers for this suggestion; however, the complexes observed in our mass spectra are very diverse and they differ their ionisation propensities. Accordingly, the relative intensities of the different species are not comparable. For instance, the signal intensities observed for the monomeric proteins are always higher than those observed for assembled complexes even if their concentrations are comparable or even lower. Likewise, signal intensities observed for the various assembled complexes also differ at similar concentrations. We, therefore, feel that a bar representation of the intensities is misleading for the unexperienced reader. Please also note that bar diagrams are not supported in the journal publishing policies.

14. I would suggest a broader discussion of how observations such as oligomerization and apparent incomplete binding may be due to the use of soluble forms in the absence of lipids - what is known about the effect lipids might have on these interactions?

We thank the reviewers for raising this important point. In general, formation of the SNARE helix bundle is unlikely to be affected by the absence of a lipid membrane as demonstrated by numerous structural studies. Nonetheless, the pre-synaptic and vesicular membranes represent the natural environment of the SNARE proteins and should be discussed.

Accordingly, previous studies discussed the possibility that the presence of a lipid membrane in *in vitro* studies might prevent SNARE complex formation by binding the hydrophobic SNARE motif. Note that the vesicle and presynaptic membranes are crowded with proteins making this scenario impossible in the natural environment of the SNARE proteins; however, this important aspect must be considered when performing *in vitro* studies without additional regulatory protein factors. Previous studies, therefore, studied membrane interactions of SNARE proteins in membrane mimetics and found that the SNARE motif is largely unstructured and only the positively charged juxtamembrane domains of Synaptobrevin-2 and Syntaxin-1A form helical segments presumably stabilising the proteins in the membranes and positioning the unstructured SNARE motifs for SNARE zippering.

We now discuss these considerations in our revised manuscript (l. 737-757; see below). However, it is important to note that the focus of our study is not on the presence of the membrane but rather the potential complexes (i.e. on- and off-pathway) that form.

“In their natural environment, the three SNAREs are anchored to the vesicular (Synaptobrevin-2) and the pre-synaptic (Syntaxin-1A and SNAP25) membranes. Employing soluble SNARE constructs in this study allows the unrestricted formation of sub-complexes in various orientations including antiparallel off-pathway complexes. Future studies, structurally characterising potential off-pathway complexes, will provide detailed insights into the mechanism and specificity of SNARE complex formation. Employing full-length proteins reconstituted in membrane mimetics will then restrict the rotational freedom of the proteins in solution and mimic the natural protein environment. However, due to lipid clustering and the comparably prevalent ionisation of lipids, the use of liposomes or nanodiscs complicates the analysis of protein complexes by native MS⁶⁸⁻⁷¹ particularly of heterogeneous mixtures of protein complexes as observed here. Continuous methodological advances are therefore required. Note that the effects of lipid membranes onto individual SNARE proteins were studied previously showing that the SNARE motifs remain mostly unstructured and only positively charged residues of the juxtamembrane regions form helical segments⁷²⁻⁷⁴. These helical segments likely stabilise the proteins in the membrane and position the unstructured SNARE motifs for SNARE zippering⁷⁵. Nonetheless, studying the interactions of SNARE proteins *in vivo* will provide valuable insights into their interactions as well as unknown intermediates in future studies. The focus of this study, however, is the stoichiometry of potential intermediates and off-pathway

complexes. Our results, therefore, significantly contribute to the understanding of SNARE complex assembly and provide the basis for future applications in native-like environments or even *in vivo*."

15. Regarding "kinetically trapped species", have you compared short with long incubation times for some of these complexes to see if reorganisation is needed and slow? Does the order in which components are added to each other matter?

As stated in our manuscript, we compared different incubation times of the different complexes. Some complexes formed immediately (for instance, the SNARE complex formed immediately after addition of Synaptobrevin-2 to the pre-assembled Syntaxin-1A:SNAP25 complex), while other formed over time. We found that the Synaptobrevin-2:SNAP25:Complexin-1 complex rearranged into the SNARE:Complexin-1 complex on a significantly longer time scale; this complex was therefore followed in more detail (see **Supplementary Figure 5**).

In this study, we also explored all different compositions of the SNARE proteins including different order of mixing (e.g. we studied the assembly of the Synaptobrevin-2:SNAP25 complex and added Syntaxin-1A as well as formation of the Syntaxin-1A:SNAP25 complex and added Synaptobrevin-2).

See manuscript for details, these experiments were described for each individual experiment in the first-submitted manuscript.

16. Volumes should be reported using L (not l)

We corrected this throughout our revised manuscript.

Reviewer #3 (Remarks to the Author):

The paper is a comprehensive analysis of neuronal SNARE protein complexes in the presence and absence of the regulatory protein Complexin by cross-linking and native mass spectrometry. Using these methods, the authors have identified and characterised several intermediate SNARE complexes as well as studied the role of Complexin on stabilising the SNARE complex. The work reveals the complexities of SNARE complex assembly that can be followed up by further analysis by other methods or in a native environment.

SNARE mediated membrane fusion is a complex and highly regulated process. The manuscript highlights the ability of using proteomics to complement structural and other biophysical methods to tease apart the multiple interactions that drive vesicle fusion. They have also shown its utility in mapping out interactions within flexible regions that are not easily resolved by structural techniques.

17. The biological relevance of the intermediate complexes are somewhat ambiguous given that soluble proteins are used rather than those in a native membrane environment. The manuscript will benefit from a short discussion of this point.

We agree with the reviewers and expanded the discussion accordingly. See also **comment #14** raised by reviewer 2 for details.

18. With the modelling some parts of the proteins were added manually. Have the authors considered using alphafold multimer to model the SNARE-Complexin complex and assessing how well their crosslinks agree with the model?

We thank the reviewers for their suggestion. We have now used ColabFold, which is implementing homology searching and AlphaFold2 (and RoseTTaFold). We used ColabFold to confirm (i) antiparallel multimerization of the SNARE complex and (ii) our model of the SNARE:Complexin-1 complex. Our models and the structural predictions of ColabFold agree very well and we have implemented this confirmation in our revised manuscript. However, as expected, AlphaFold(2) is limited when concerning unstructured (disordered) regions. As there are no high-resolution templates for these proteins (or protein regions), they cannot be fully integrated in the models. We experienced this mostly for Complexin-1, for which no full-length high-resolution structures are available. The disordered regions or Complexin-1 are, however, of utmost importance during SNARE binding and SNARE regulation. Our

manual refinement of the model therefore outperforms the predicted model. We modified the text accordingly at various places (l. 484-506, l. 536-548, l. 7314-736). Alongm the same line, **Supplementary Figures 11** and **12** as well as **Figure 6** were modified or created:

Supplementary Figure 11

Supplementary Figure 12

Figure 6

19. How do the authors reconcile their observations with the fast kinetics associated with vesicle fusion?

In principle, membrane fusion can successfully be performed *in vitro* using purified SNARE proteins. However, the main difference between fusion experiments performed in *in vitro* assays and occurring in the natural cell environment are the fast kinetics of the natural system. These fast kinetics mostly originate from fine tuning by regulatory proteins factors or other factors such as post-translational modifications in the natural environment. Regulatory proteins such as Munc18, Munc13, Complexin, Synaptotagmin-1 and others maintain partially assembled SNARE complexes, which are activated upon Ca-influx. These conditions cannot be reflected in *in vitro* assays such as those presented in this study; nonetheless, the on- and off-pathway intermediates observed in this study will help our further understanding of the assembly process of the SNARE complex. The kinetic differences observed between the various complexes do not reflect differences in the ability of the complexes to perform membrane fusion but rather provide clues on their structural arrangements.

20. Further validation of their observations through mutational analysis will be beneficial and perhaps warrants a comment in the discussion.

We agree with the reviewers that mutational analysis is always beneficial when studying protein interactions. We added the following comment to the discussion section (I. 735-739), in which we discuss the advantages of native mass spectrometry:

“Mutational analysis introducing point mutations at key interaction sites as well as deletion of structural elements will provide additional insights in future experiments.”

REVIEWERS' COMMENTS:

Reviewer #1 (Remarks to the Author):

Reviewer's comments on the revised manuscript by Hesselbarth and Schmidt "A Roadmap for SNARE Complex Assembly: Mass Spectrometry Uncovers Protein Interactions and Stoichiometry of Intermediates and Sub-complexes"

In their revised manuscript, Hesselbarth and Schmidt have addressed the points I made on their original version in a satisfying manner.

I can therefore recommend publishing the manuscript in Communications Biology.

Reviewer #2 (Remarks to the Author):

I would like to thank the authors that they addressed my comments, I believe that this paper is now ready for publication.

Reviewer #3 (Remarks to the Author):

In this revised manuscript the researchers have taken the time to thoroughly address all of the questions and comments from the reviewers.

The manuscript now explains more clearly the characterisation and significance of the different SNARE protein species identified by mass spectrometry, while demonstrating at the same time the utility and limitations of the method for the analysis of protein complexes. As such this body of work will be of great value to those in the membrane trafficking fields as well as those involved in studying protein complexes in general.